# MAKING OFFLINE MODEL-BASED REINFORCEMENT LEARNING WORK ON REAL ROBOTS

## ABSTRACT

Reinforcement Learning (RL) has achieved impressive results in robotics, yet high-performing pipelines remain highly task-specific, with little reuse of prior data. Offline Model-based RL (MBRL) offers greater data efficiency by training policies entirely from existing datasets, but suffers from compounding errors and distribution shift in long-horizon rollouts. Although existing methods have shown success in controlled simulation benchmarks, robustly applying them to the noisy, biased, and partially observed datasets typical of real-world robotics remains challenging. We present a principled pipeline for making offline MBRL effective on physical robots. Our RWM-O extends autoregressive world models with epistemic uncertainty estimation, enabling temporally consistent multi-step rollouts with uncertainty effectively propagated over long horizons. We combine RWM-O with MOPO-PPO, which adapts uncertainty-penalized policy optimization to the stable, on-policy PPO framework for real-world control. We evaluate our approach on diverse manipulation and locomotion tasks in simulation and on a real quadruped, training policies entirely from offline datasets. The resulting policies consistently outperform model-free and uncertainty-unaware model-based baselines, and fusing real-world data in model learning further yields robust policies that surpass online model-free baselines trained solely in simulation.

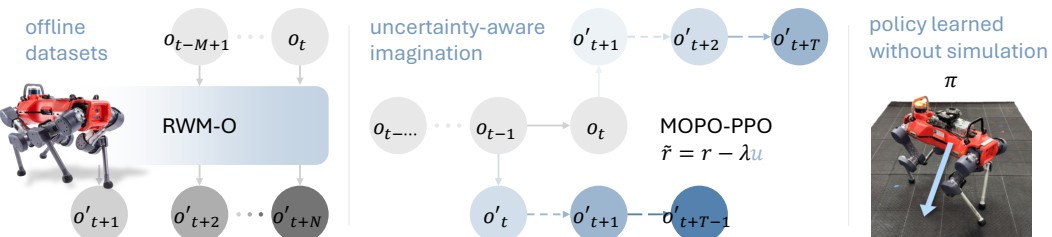

Figure 1: Overview of RWM-O and MOPO-PPO. RWM-O is trained on offline data to predict long-horizon dynamics with an ensemble-based uncertainty estimate. The shade visualizes epistemic uncertainty, which is penalized during policy optimization to avoid overfitting to unreliable predictions. Actions are omitted for clarity. Our framework enables direct policy learning without simulators, bridging the gap between model-based offline RL and real-world robotic deployment.

## 1    INTRODUCTION

Reinforcement Learning (RL) has achieved impressive results in robotic control across manipulation and locomotion tasks (Miki et al., 2022; Li et al., 2023b;a; Hoeller et al., 2024). While state-of-the-art model-free RL can train policies with access to high-fidelity physics simulators that transfer effectively to real robots, these training pipelines are typically highly task-specific: rewards, environments, and policy tuning are tailored for each individual task (Peng et al., 2018; Hwangbo et al., 2019; Rudin et al., 2022; Mittal et al., 2023). As a result, the data and models developed for one setup rarely share information with others, even when the underlying dynamics or skills are related (Cobbe et al., 2019; Bengio et al., 2020). Data reuse in robotics requires algorithms that can extract value from large, heterogeneous datasets gathered across different tasks, robots, and simulators. *Offline* RL addresses

this gap by learning policies entirely from existing datasets, collected in simulation or on hardware, without requiring additional environment interactions (Deisenroth & Rasmussen, 2011; Chua et al., 2018; Clavera et al., 2018; Yu et al., 2020). By leveraging task- and policy-agnostic datasets, it can produce policies that not only outperform the data-collecting policies but also be repurposed for entirely different tasks. This ability to exploit broad, previously gathered data makes offline RL a promising framework for improving efficiency and flexibility in robotics.

Despite this promise, offline RL has seen limited adoption in real-world robotics (Zhou et al., 2022; Li et al., 2023c) (see Sec. A.5). Static datasets are often biased toward the behavior policy that generated them and cover a narrow subset of the state-action space (Chua et al., 2018; Clavera et al., 2018; Deisenroth & Rasmussen, 2011). These issues lead to distribution shift, where deployed policies encounter states outside the dataset's support, making predictions unreliable. Some model-free offline RL methods mitigate this by enforcing strict constraints to keep the policy within the dataset's support, but at the cost of limiting generalization beyond observed transitions (Agarwal et al., 2019; Kumar et al., 2019; Wu et al., 2019). In contrast, model-based RL (MBRL) tackles the problem by learning a predictive dynamics model and applying techniques such as uncertainty-aware modeling and uncertainty-penalized policy optimization to reduce the risk of unreliable predictions (Gal et al., 2016; Lakshminarayanan et al., 2017; Kuleshov et al., 2018; Ovadia et al., 2019; Yu et al., 2020). While these methods have shown success in controlled simulation benchmarks, their reliable deployment in real robotics remains challenging, as noisy, biased, and partially observed datasets demand both accurate long-horizon modeling and stable policy learning.

In this work, we present a principled pipeline for making model-based offline RL effective in real-world robotic settings. Our Offline Robotic World Model (RWM-O) extends dynamics modeling with autoregressive long-horizon prediction and ensemble-based epistemic uncertainty estimation, enabling temporally consistent multi-step rollouts. We integrate RWM-O with MOPO-PPO, an adaptation of MOPO's uncertainty-penalized optimization to PPO, which has been widely adopted in modern real-world robotic control (Yu et al., 2020). Unlike MOPO's one-step setup, our method must propagate and manage uncertainty over 100-step episodic rollouts, controlling compounding model errors without any further data collection. We validate our pipeline on diverse simulated tasks and real quadruped locomotion, where policies are trained entirely offline from existing datasets and successfully deployed on hardware. We show that fusing real-world data in model learning yields robust policies that surpass state-of-the-art online model-free baselines trained solely in simulation. To our best knowledge, this is the first demonstration of uncertainty-penalized offline MBRL operating full-scale tasks on a physical robot. Our results show that with the right integration of modeling, uncertainty handling, and policy optimization, offline RL can be a practical framework for reusing past robotic experience—effectively reusing both simulation logs and real-world data across different tasks rather than treating each development as entirely isolated. Supplementary videos for this work are available at `https://sites.google.com/view/iclr2026-rwm-o/home`.

## 2 RELATED WORK

### 2.1 MODEL-BASED REINFORCEMENT LEARNING

Model-Based Reinforcement Learning (MBRL) constructs an explicit predictive model of environment dynamics, improving sample efficiency and facilitating long-horizon planning. Unlike model-free RL, which directly learns policies from data, MBRL generates synthetic rollouts to optimize policies, reducing reliance on real-world interactions. Early approaches such as PILCO leverage Gaussian processes to model system dynamics, achieving data-efficient learning but facing scalability issues in high-dimensional spaces (Deisenroth & Rasmussen, 2011). Local linear models, as seen in guided policy search (Levine & Koltun, 2013), improve tractability in complex control tasks (Kumar et al., 2016). Neural network-based dynamics models have since become the dominant approach, allowing for greater expressivity and scalability (Ebert et al., 2018; Kaiser et al., 2019). One notable example is Model-Based Policy Optimization (MBPO), which interleaves model-generated rollouts with model-free updates, selectively trusting the learned model to improve stability (Janner et al., 2019). Recent advancements in latent-space world models have further improved MBRL. PlaNet introduces a compact latent dynamics model for planning directly in an abstract space, making it computationally feasible for long-horizon tasks (Hafner et al., 2019b). Dreamer extends this idea by integrating an actor-critic framework, achieving strong performance in continuous control (Hafner

et al., 2019a; 2020; 2023). Robotic World Model (RWM) introduces a self-supervised, autoregressive neural simulator that learns long-horizon environment dynamics without domain-specific inductive biases (Li et al., 2025). While these methods perform well in online settings, where model errors can be corrected through additional environment interactions, their applicability to offline settings remains underexplored.

## 2.2 UNCERTAINTY ESTIMATION AND DISTRIBUTION SHIFT MITIGATION

Incorporating uncertainty estimation into MBRL is crucial, especially in offline settings where the lack of online interactions limits the ability to correct model inaccuracies. One of the most effective and widely used techniques for model uncertainty estimation is the use of bootstrap ensembles, which have been successfully applied in various MBRL frameworks to mitigate overestimation caused by distribution shifts in offline RL (Lakshminarayanan et al., 2017; Chua et al., 2018; Yu et al., 2020). Probabilistic Ensembles with Trajectory Sampling (PETS) pioneers the use of probabilistic ensembles in MBRL, improving robustness by leveraging multiple network predictions for decision-making (Chua et al., 2018). To further improve uncertainty estimation and its integration into policy learning, several advanced techniques have been developed. MOUP introduces ensemble dropout networks for uncertainty estimation while incorporating a maximum mean discrepancy constraint into policy optimization, ensuring bounded state mismatch and improving policy learning in offline settings (Zhu et al., 2024). SUMO characterizes uncertainty by measuring the cross-entropy between model dynamics and true dynamics using a k-nearest neighbor search method (Qiao et al., 2024). This technique provides an alternative perspective on estimating uncertainty beyond conventional model ensembles. Uncertainty-aware policy optimization is key to mitigating the risks of distribution shift. MORPO explicitly quantifies the uncertainty of predicted dynamics and incorporates a reward penalty to balance optimism and conservatism in policy learning (Guo et al., 2022). Similarly, CQL (Kumar et al., 2020), IQL (Kostrikov et al., 2021), and COMBO (Yu et al., 2021) tackle uncertainty estimation challenges by enforcing conservatism in value estimation without relying on explicit uncertainty quantification. While these methods achieve impressive performance in controlled simulation benchmarks, applying them to real-world robotics remains a significant hurdle, where reliability and robustness demand both accurate long-horizon modeling and stable policy learning. We highlight some of these real challenges in Sec. A.5.

## 3 PRELIMINARIES

### 3.1 OFFLINE MODEL-BASED REINFORCEMENT LEARNING

We formulate the problem by modeling the environment as a Partially Observable Markov Decision Process (POMDP) (Sutton & Barto, 2018), defined by the tuple $(\mathcal{S}, \mathcal{A}, \mathcal{O}, T, R, O, \gamma)$, where $\mathcal{S}$, $\mathcal{A}$, and $\mathcal{O}$ denote the state, action, and observation spaces, respectively. The transition kernel $T : \mathcal{S} \times \mathcal{A} \to \mathcal{S}$ captures the environment dynamics, while the reward function $R : \mathcal{S} \times \mathcal{A} \times \mathcal{S} \to \mathbb{R}$ maps transitions to scalar rewards. The agent seeks to learn a policy $\pi_\theta : \mathcal{O} \to \mathcal{A}$ that maximizes the expected discounted return $\mathbb{E}_{\pi_\theta} \left[ \sum_{t \geq 0} \gamma^t r_t \right]$, where $r_t$ is the reward at time $t$ and $\gamma \in [0, 1]$ is the discount factor.

In offline RL, the agent is trained using a fixed dataset of transitions collected by one or a mixture of behavior policies. Unlike online RL, the agent cannot interact with the environment, making it susceptible to extrapolation errors when encountering out-of-distribution states (Fujimoto et al., 2019; Fu et al., 2020). Many offline RL methods impose conservatism by constraining the policy to remain within the dataset's support, limiting generalization beyond observed trajectories (Kumar et al., 2019; Wu et al., 2019). MBRL provides an alternative approach by learning a world model of the environment, which can be leveraged for policy optimization in *imagination* (Sutton, 1991). Instead of relying solely on offline data for policy learning, MBRL methods construct a predictive model of the transition dynamics, allowing the agent to generate synthetic rollouts and expand the effective training data. This enables more efficient learning and better generalization, which is particularly important in offline settings (Yu et al., 2020). However, model errors can accumulate over long-horizon predictions, introducing bias and degrading policy performance. Addressing these challenges requires accurate long-horizon modeling and robust handling of model errors.

## 3.2 ROBOTIC WORLD MODEL

Robotic World Model (RWM) is a neural network-based world model designed to accurately capture long-horizon robotic dynamics without requiring domain-specific inductive biases (Li et al., 2025). Unlike conventional methods that rely on structured priors or handcrafted features, RWM generalizes across diverse robotic systems using a self-supervised, dual-autoregressive training framework. RWM learns to predict future observations autoregressively, feeding its own predictions back into the model to simulate long-horizon rollouts. The training objective minimizes a multi-step prediction loss

$$\mathcal{L} = \frac{1}{N} \sum_{k=1}^{N} \alpha^k \left[ L_o \left( o'_{t+k}, o_{t+k} \right) \right], \tag{1}$$

where $N$ denotes the autoregressive prediction steps, $o'_{t+k}$ is the predicted observations, $L_o$ measures the prediction discrepancy, and $\alpha_k$ is a decay factor. The predicted observation $k$ steps ahead of time $t$ can be written as

$$o'_{t+k} \sim p_\phi \left( \cdot \mid o_{t-M+k:t}, o'_{t+1:t+k-1}, a_{t-M+k:t+k-1} \right), \tag{2}$$

where $M$ denotes the history horizon steps. Through this design, RWM achieves robust trajectory forecasting across manipulation and locomotion tasks. Policies trained with RWM have been zero-shot deployed on hardware, achieving high-fidelity control without requiring additional fine-tuning. However, RWM lacks uncertainty estimation, requiring policy training to be conducted *online*, where additional interactions induced by the current policy are leveraged to correct model errors, making this approach infeasible in *offline* settings.

## 4 APPROACH

### 4.1 UNCERTAINTY QUANTIFICATION WITH OFFLINE ROBOTIC WORLD MODEL

In MBRL, the learned dynamics model inevitably becomes inaccurate when making predictions for state-action pairs that deviate from the distribution of the training data. This issue is particularly critical in the offline setting, where errors in the learned dynamics cannot be corrected through additional environment interactions. As a result, standard model-based policy optimization methods risk overfitting to erroneous model predictions in out-of-distribution regions, leading to suboptimal or unsafe policy behavior. To ensure reliable performance, it is essential to balance the trade-off between policy improvement and model reliability: leveraging model-based imagination to explore high-return policies beyond the support of the dataset while mitigating the risk of policy degradation due to overfitting to model errors in regions with high uncertainty.

To this end, we introduce Offline Robotic World Model (RWM-O), where we explicitly incorporate uncertainty quantification into the dynamics model based on RWM. We aim to design an uncertainty estimator that captures both *epistemic* uncertainty, which arises due to limited training data, and *aleatoric* uncertainty, which reflects inherent stochasticity in the environment. Bootstrap ensembles have been shown to provide consistent estimates of uncertainty (Bickel & Freedman, 1981) and have been successfully applied in MBRL to improve robustness (Chua et al., 2018). RWM-O extends RWM by introducing an ensemble-based uncertainty-aware architecture. Specifically, we apply bootstrap ensembles to the observation prediction head of the model after a shared recurrent feature extractor (e.g., a GRU). Each ensemble element $b$ independently predicts the mean and variance of a Gaussian distribution over the next observation

$$o'^b_{t+k} \sim \mathcal{N} \left( \mu^b_{o_{t+k}}, \sigma^{2,b}_{o_{t+k}} \right), \tag{3}$$

where $\mu^b_{o_{t+k}}, \sigma^{2,b}_{o_{t+k}}$ denote the mean and variance predicted by ensemble member $b$, following the autoregressive prediction scheme in Eq. 2. The learned variance component captures aleatoric uncertainty effectively. The training objective is then computed as the average loss over all ensemble members, following Eq. 1.

During inference, the ensemble mean is used as the predicted next observation, while the variance across ensemble members provides an estimate of epistemic uncertainty $u_{p_\phi}$, depending on the world model $p_\phi$. The training overview is visualized in Fig. 1.

$$o'_{t+1} = \mathbb{E}_b \left[ \mu^b_{o_{t+1}} \right], u_{t+1} = u_{p_\phi} \left( o_{t-M+1:t}, a_{t-M+k:t} \right) = \mathrm{Var}_b \left[ \mu^b_{o_{t+1}} \right]. \tag{4}$$

By incorporating uncertainty-aware modeling into the dynamics learning process, RWM-O enables robust trajectory forecasting in offline settings with uncertainty effectively propagated over long horizons. The explicit quantification of epistemic uncertainty allows for uncertainty-informed policy optimization, preventing the policy from over-relying on uncertain model predictions.

### 4.2 POLICY OPTIMIZATION ON UNCERTAINTY-PENALIZED IMAGINATION

To ensure robust policy learning in offline MBRL, we introduce Model-Based Offline Policy Optimization with Proximal Policy Optimization (MOPO-PPO), an adaptation of the MOPO framework (Yu et al., 2020) to PPO (Schulman et al., 2017). MOPO-PPO allows cautious exploration beyond the behavioral distribution while penalizing transitions with high epistemic uncertainty. Specifically, given the uncertainty estimator $u_{p_\phi}$ from Eq. 4, we modify the reward function to discourage the policy from exploiting unreliable model predictions:

$$\tilde{r}\left(o_t, a_t\right) = r_t\left(o_t, a_t\right) - \lambda u_{p_\phi}\left(o_{t-M+1:t}, a_{t-M+k:t}\right), \tag{5}$$

where $\lambda$ is a hyperparameter controlling the penalty strength. This formulation encourages the policy to focus on high-confidence regions of the learned dynamics model while avoiding overfitting to model errors.

The training process of MOPO-PPO using RWM-O is outlined in Fig. 1 and Algorithm 1.

---

**Algorithm 1** Offline policy optimization with RWM-O

---

1: **Input:** Offline dataset $\mathcal{D}$
2: Initialize policy $\pi_\theta$, world model $p_\phi$
3: Train the world model $p_\phi$ autoregressively using offline dataset $\mathcal{D}$ according to Eq. 1
4: **for** learning iterations $= 1, 2, \ldots$ **do**
5:     Initialize imagination agents with observations sampled from $\mathcal{D}$
6:     Roll out uncertainty-aware trajectories using $\pi_\theta$ and $p_\phi$ for $T$ steps according to Eq. 4
7:     Compute uncertainty-penalized rewards for each imagination transition according to Eq. 5
8:     Update $\pi_\theta$ using PPO or another RL algorithm
9: **end for**

---

By integrating uncertainty-aware dynamics modeling with policy optimization, RWM-O and MOPO-PPO enable safer and more reliable offline MBRL. Explicit uncertainty estimation mitigates distribution shift, allowing cautious generalization beyond the dataset while ensuring stability through trust-region optimization. By leveraging uncertainty-aware rollouts, MOPO-PPO enhances policy robustness and efficiency, advancing scalable offline RL for real-world robotics.

## 5 EXPERIMENTS

We evaluate RWM-O and MOPO-PPO through a comprehensive set of experiments spanning diverse simulated and real robotic platforms, tasks, and dataset sources. Our objective is to assess the capability of RWM-O in quantifying uncertainty during long-horizon rollouts and to demonstrate the effectiveness of MOPO-PPO in improving policy performance through uncertainty-aware optimization in *real robotic settings*. We detail our distinct challenges in Sec. A.5.

### 5.1 AUTOREGRESSIVE UNCERTAINTY ESTIMATION

A reliable estimate of epistemic uncertainty is critical for ensuring robust policy learning in offline MBRL. In particular, the ability of a world model to quantify its own uncertainty over long-horizon rollouts determines the trustworthiness of its synthetic experience. To evaluate this property in RWM-O, we analyze the alignment between its predicted epistemic uncertainty and actual model prediction errors during autoregressive trajectory forecasting on ANYmal D. The observation and action spaces of the world model are detailed in Table S2 and Table S4.

We train the world model on offline data collected from a velocity-tracking policy, operating at a control frequency of $50\,Hz$. The model is trained with a history horizon of $M = 32$ and a prediction horizon of $N = 8$, using an ensemble of five networks to estimate epistemic uncertainty. To evaluate

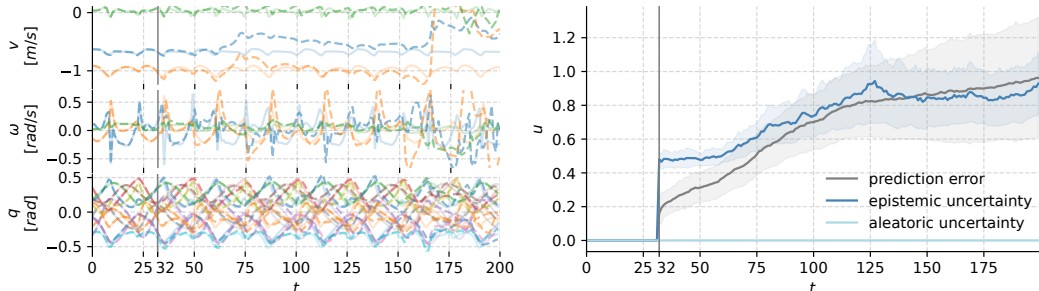

Figure 2: Autoregressive trajectory prediction (left) and uncertainty estimation (right) by RWM-O. Solid lines represent ground truth, while dashed lines denote predicted state evolution. Predictions commence at $t = 32$ using historical observations, with future observations predicted autoregressively by feeding prior predictions back into the model. The epistemic uncertainty estimate by RWM-O aligns with the long-horizon prediction error and thus sets a reliable metric in policy training.

uncertainty estimation, we compute epistemic uncertainty as the variance of the ensemble predictions, following Eq. 4. Aleatoric uncertainty is estimated as the mean of the predicted standard deviations across ensemble members. Details of the network architecture and training setup are provided in Sec. A.2.1 and Sec. A.3.1. The predictive error, epistemic uncertainty, and aleatoric uncertainty over autoregressive rollouts are visualized in Fig. 2.

At $t = 32$, the model transitions from conditioning on ground-truth observations to fully autoregressive rollouts, wherein predicted states are recursively fed back into the model. As expected, the accumulated prediction error (grey) increases over time due to compounding model inaccuracies. Importantly, the estimated epistemic uncertainty (dark blue) closely follows the trend of the prediction error, demonstrating that RWM-O effectively captures uncertainty in regions where the model generalization deteriorates. In contrast, the aleatoric uncertainty (light blue) remains low, reflecting small stochasticity in the environment. The strong correlation between epistemic uncertainty and model prediction error justifies its role as a trust metric for policy optimization. By penalizing high-uncertainty transitions, MOPO-PPO prevents policy exploitation in poorly modeled regions of the state space, mitigating the risk of compounding errors in long-horizon rollouts. These results validate the proposed uncertainty-aware regularization strategy, reinforcing the efficacy of RWM-O in enabling robust policy learning in offline RL.

## 5.2 UNCERTAINTY-PENALIZED IMAGINATION

To evaluate the role of uncertainty-aware regularization in policy learning, we train RWM-O on an offline dataset collected from an expert velocity-tracking policy on ANYmal D. We then optimize policies with MOPO-PPO using different values of the uncertainty penalty coefficient $\lambda$ in Eq. 5, which is a hyperparameter that controls the trade-off between exploration and model reliability. Figure 3 shows the evolution of the *imagination* reward and epistemic uncertainty over training iterations, as well as the final policy *evaluation* performance.

When the penalty is small ($\lambda \leq 0.5$), the policy attains high rewards in imagination but exploits model inaccuracies, leading to overfitting to hallucinated rollouts. This results in poor evaluation performance, as the learned policy exhibits overconfident behaviors in simulation but fails catastrophically when tested on real dynamics. In these cases, the robot often struggles to maintain stable locomotion, exhibiting unintended collisions or failing to walk altogether. In contrast, large penalties ($\lambda \geq 2.0$) force the policy to remain in low-uncertainty regions, limiting exploration and leading to overly conservative behaviors. These policies exhibit excessive caution, often resulting in the robot standing in place or making only minor, ineffective movements. Consequently, the learned policy fails to make meaningful progress, as indicated by the low final evaluation rewards.

With an appropriate penalty ($\lambda = 1.0$), the policy effectively balances exploration and exploitation. The imagination reward increases steadily, while epistemic uncertainty remains controlled, allowing the policy to leverage synthetic rollouts without overfitting to model errors. This results in effective learning, where the robot is able to explore new behaviors in imagination while remaining in

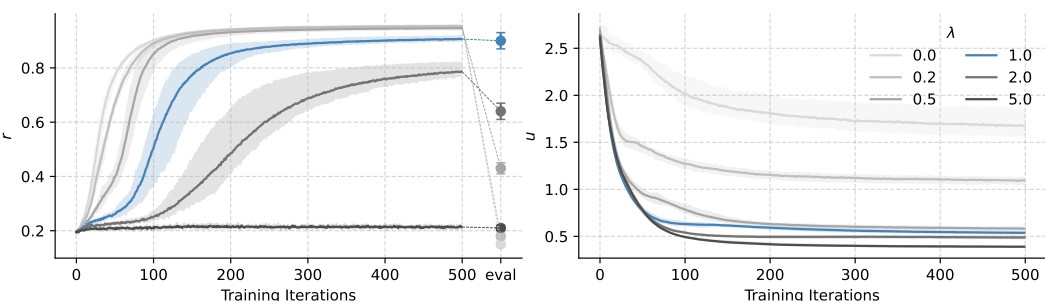

Figure 3: Imagination reward (left) and epistemic uncertainty (right) during MOPO-PPO training. The policy evaluation on *real* dynamics is visualized in dots (left). Small penalties lead to overconfident but unreliable policies, while large penalties result in overly conservative behaviors. A well-calibrated penalty (dark blue) achieves the exploration-exploitation balance.

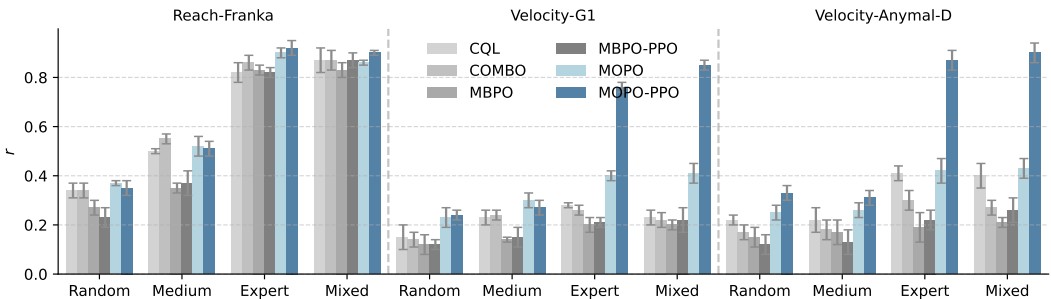

Figure 4: Normalized episodic rewards across diverse robotic environments and offline RL algorithms with different dataset types. MOPO-PPO consistently outperforms uncertainty-unaware baselines, particularly in complex locomotion tasks.

trustworthy regions of the learned model. The strong alignment between training progress and final evaluation performance highlights the effectiveness of MOPO-PPO in leveraging epistemic uncertainty to mitigate distribution shift, ultimately enhancing policy robustness in offline MBRL.

## 5.3 GENERALITY ACROSS ROBOTIC ENVIRONMENTS

To evaluate the generality and robustness of MOPO-PPO with RWM-O across diverse robotic environments, we compare its performance against widely adopted offline RL baselines, including Conservative Q-Learning (CQL) (Kumar et al., 2020), COMBO (Yu et al., 2021), MBPO (Janner et al., 2019), MBPO-PPO (Li et al., 2025), and MOPO (Yu et al., 2020). These baselines represent state-of-the-art approaches in model-free and model-based offline RL. For a fair comparison, we train MBPO and MBPO-PPO in an offline setting, treating them as uncertainty-unaware counterparts to MOPO and MOPO-PPO as detailed in Sec. A.2.3.

We benchmark these methods on three robotic tasks in simulation: *Reach-Franka* (manipulation), *Velocity-G1* (humanoid locomotion), and *Velocity-ANYmal-D* (quadruped locomotion), spanning a broad range of dynamics complexities and control challenges. Each method is trained on four dataset types: **Random**: trajectories collected from a randomly initialized policy, **Medium**: trajectories from a partially trained PPO policy, **Expert**: trajectories from a fully trained PPO policy, **Mixed**: a combination of replay buffer data from PPO training at various stages. All methods are trained and evaluated under identical conditions. The normalized episodic rewards for each method across different environments and dataset types are presented in Fig. 4.

Across all methods, performance improves with increasing dataset quality, as richer data distributions reduce the impact of distribution shift. Policies trained on the **Mixed** dataset achieve the highest rewards due to its broad state-action coverage, comparable to those trained on **Expert** data, which tends to overfit to specific dynamics and biases the policy towards suboptimal generalization.

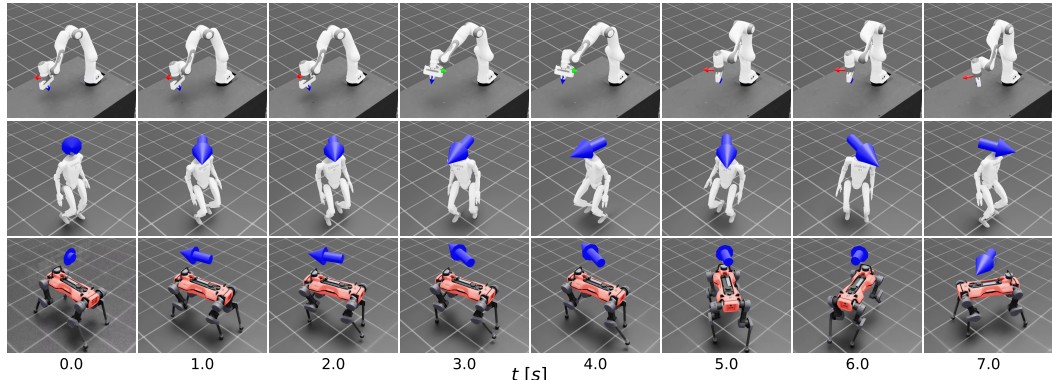

Figure 5: MOPO-PPO policy deployment learned from offline **Mixed** dataset across various environments. Coordinate and arrow markers denote the end-effector pose and base velocity, respectively.

While CQL, COMBO, and SAC-based model-based methods perform competitively in the manipulation task, they demonstrate significantly weaker performance in locomotion tasks, where the complexity of dynamic interactions amplifies the limitations of value-based learning. As highlighted in Sec. 5.2, MBPO and MBPO-PPO, which lack explicit uncertainty estimation, tend to overfit to model inaccuracies, leading to overconfident policies. In manipulation tasks, this overfitting is mitigated when dataset coverage is sufficient. However, in locomotion tasks, it results in catastrophic failures, including robot collisions and falls.

In contrast, MOPO-PPO consistently outperforms baselines across all tasks, particularly in *Velocity-G1* and *Velocity-ANYmal-D*, where uncertainty-aware rollouts significantly enhance policy stability. By penalizing high-uncertainty transitions, MOPO-PPO prevents policies from exploiting hallucinated model predictions, leading to safer and more reliable locomotion behaviors. The deployment of the learned policy is visualized in Fig. 5 with supplementary videos available on our webpage. The strong performance of MOPO-PPO across different dataset types and environments demonstrates its effectiveness in mitigating distribution shift, reinforcing its applicability to real-world robotic control.

### 5.4 OFFLINE LEARNING WITH REAL-WORLD DATA

To validate the real-world applicability of RWM-O and MOPO-PPO, we train a quadruped locomotion policy entirely from offline datasets that combine different proportions of simulation and real-world data. For simulation, we use the **Mixed** dataset collected from a suboptimal velocity-tracking policy under heavy randomization. For real-world data, we deploy the same suboptimal policy and collect trajectories using an autonomous pipeline with minimal human intervention, as described in Sec. A.4. Table 1 reports the normalized real-world policy performance, where the data-collecting policy achieves an average score of $0.79$. Remarkably, MOPO-PPO surpasses this baseline even when trained exclusively on simulated transitions, highlighting the ability of offline MBRL to distill stronger policies than those present in the dataset. Incorporating a moderate amount of real-world data further improves performance by reducing domain shift, yielding the best result of $0.91 \pm 0.03$ with 800K simulated and 200K real transitions, surpassing the policies trained with online model-free methods purely in simulation. However, excessive reliance on real-world data reduces performance and increases failure rates, reflecting the limited diversity of real trajectories that lack informative but risky behaviors (e.g., failure modes) necessary for robust world model and policy learning.

Table 1: Normalized policy performance with different data mixtures

| Sim | 1M | 800K | 600K | 400K | collecting | online model- |
|------|------|------|------|------|------|------|
| Real | 0 | 200K | 400K | 600K | policy | free policy |
| Perf. | $0.82 \pm 0.02$ | $\mathbf{0.91 \pm 0.03}$ | $0.78 \pm 0.02$ | $0.59 \pm 0.04$ | $0.79 \pm 0.02$ | $0.88 \pm 0.01$ |

We additionally study the effect of uncertainty penalization during policy training (Fig. 6). With insufficient penalization ($\lambda = 0.2$), the robot overfits to model errors, producing unstable gaits and

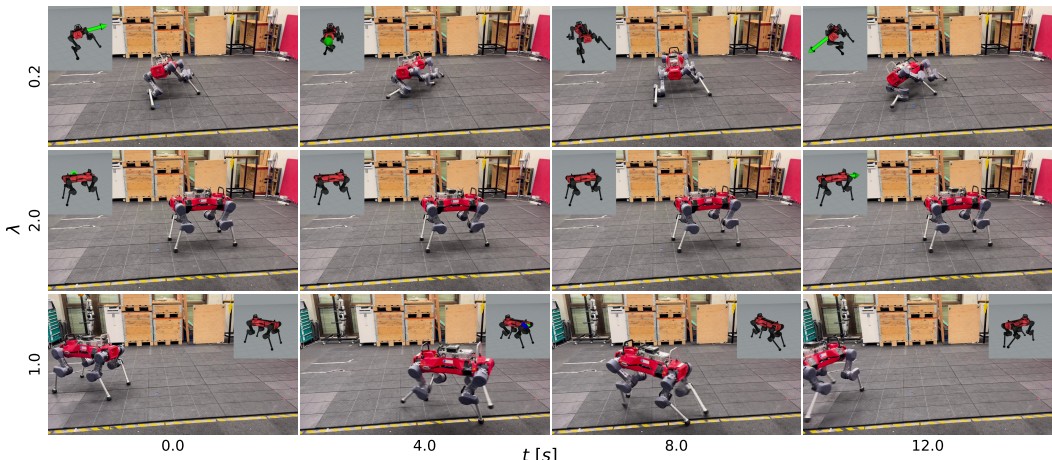

Figure 6: MOPO-PPO policy deployment on ANYmal D under different levels of model uncertainty penalization. The rows show policy performance with insufficient ($\lambda = 0.2$), excessive ($\lambda = 2.0$) and appropriate ($\lambda = 1.0$) model uncertainty penalization during policy training.

frequent collisions despite high imagined rewards. Conversely, excessive penalization ($\lambda = 2.0$) constrains the policy to low-uncertainty regions, resulting in overly conservative behaviors such as standing still or minimal, ineffective movements. A well-calibrated penalty ($\lambda = 1.0$) strikes a balance: the imagination reward increases steadily while epistemic uncertainty remains controlled, enabling the robot to explore effectively in imagination while remaining in reliable regions of the learned dynamics model. This calibration leads to stable and robust locomotion on hardware. Supplementary videos demonstrating the learned policies on hardware are available on our project webpage.

## 6 LIMITATIONS

Although RWM-O and MOPO-PPO advance offline MBRL for robotics, some limitations remain. As with all offline RL approaches, policy quality is ultimately bounded by the dataset: limited state-action coverage and compounding model errors over long horizons can still degrade generalization. Our uncertainty-aware training substantially mitigates these effects but cannot eliminate them entirely. In addition, although our method enables training with offline real-world datasets, such data often lack risky but informative transitions that are essential for world model and policy learning. In practice, simulated environments remain valuable as a complementary source for generating these transitions safely. This highlights a trade-off between realism and diversity: real-world data mitigates domain shift, while simulation can enrich coverage of rare but critical experiences.

## 7 CONCLUSION

In this work, we introduce a principled pipeline for making offline MBRL effective in realistic robotic settings. Our approach, RWM-O combined with MOPO-PPO, addresses the challenges of compounding model errors and distribution shift by extending autoregressive world models with epistemic uncertainty estimation and integrating uncertainty-penalized policy optimization into a stable on-policy framework. This enables long-horizon rollouts where uncertainty is effectively propagated, allowing policies to generalize cautiously beyond the dataset while maintaining robustness to model errors. Through experiments across simulated manipulation and locomotion tasks and on a real quadruped robot, we demonstrated that our method consistently outperforms model-free and uncertainty-unaware model-based baselines, achieving, to our knowledge, the first successful deployment of offline MBRL on physical hardware. These results establish that offline RL, when combined with principled modeling and uncertainty handling, can serve as a practical framework for reusing past robotic experience.

ETHICS STATEMENT

This work does not involve human subjects or sensitive data. All experiments are conducted in simulation or on dedicated robotic hardware operated by the authors, with no use of third-party datasets. The research complies with the Code of Ethics of the venue. The proposed framework provides a robust and scalable method for learning world models tailored to complex robotic tasks. This can benefit domains such as healthcare, disaster response, and logistics, and reduce environmental and hardware costs associated with physical experimentation. Potential risks include misuse of the method in surveillance or autonomous enforcement systems, and the acceleration of automation in labor-sensitive sectors. While such uses are not intended or explored in this work, the authors acknowledge the dual-use potential of generalizable control methods. To mitigate safety risks, policy training occurs entirely in imagination, and deployment is limited to policies validated under domain shifts. Failure events are explicitly modeled and used to terminate unsafe rollouts.

REPRODUCIBILITY STATEMENT

The experiment results presented in this work can be reproduced with the implementation details provided in Sec. A.1, Sec. A.2, and Sec. A.3. The code will be open-sourced on the project page upon acceptance.

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

# A APPENDIX

## A.1 TASK REPRESENTATION

### A.1.1 OBSERVATION AND ACTION SPACES

The observation space for the ANYmal world model is composed of base linear and angular velocities $v, \omega$ in the robot frame, measurement of the gravity vector in the robot frame $g$, joint positions $q$, velocities $\dot{q}$ and torques $\tau$ as in Table S2.

Table S2: World model observation space

| Entry | Symbol | Dimensions |
|---|---|---|
| base linear velocity | $v$ | 0:3 |
| base angular velocity | $\omega$ | 3:6 |
| projected gravity | $g$ | 6:9 |
| joint positions | $q$ | 9:21 |
| joint velocities | $\dot{q}$ | 21:33 |
| joint torques | $\tau$ | 33:45 |

Some auxiliary information is used as additional regression signals to encourage learning features useful for long-term dynamics predictions. For example, such information for ANYmal world model is composed of knee and foot contacts as in Table S3.

Table S3: World model auxiliary target

| Entry | Symbol | Dimensions |
|---|---|---|
| knee contact | – | 0:4 |
| foot contact | – | 4:8 |

The action space is composed of joint position targets as in Table S4.

Table S4: Action space

| Entry | Symbol | Dimensions |
|---|---|---|
| joint position targets | $q^*$ | 0:12 |

The observation space for the ANYmal velocity tracking policy is composed of base linear and angular velocities $v, \omega$ in the robot frame, measurement of the gravity vector in the robot frame $g$, velocity command $c$, joint positions $q$ and velocities $\dot{q}$ as in Table S5.

### A.1.2 REWARD FUNCTIONS

The total reward is sum of the following terms with weights detailed in Table S6.

Linear velocity tracking $x, y$

$$r_{v_{xy}} = w_{v_{xy}} e^{-\|c_{xy} - v_{xy}\|_2^2 / \sigma_{v_{xy}}^2},$$

where $\sigma_{v_{xy}} = 0.25$ denotes a temperature factor, $c_{xy}$ and $v_{xy}$ denote the commanded and current base linear velocity.

Angular velocity tracking

$$r_{\omega_z} = w_{\omega_z} e^{-\|c_z - \omega_z\|_2^2 / \sigma_{\omega_z}^2},$$

Table S5: Policy observation space

| Entry | Symbol | Dimensions |
|-------|--------|------------|
| base linear velocity | $v$ | 0:3 |
| base angular velocity | $\omega$ | 3:6 |
| projected gravity | $g$ | 6:9 |
| velocity command | $c$ | 9:12 |
| joint positions | $q$ | 12:24 |
| joint velocities | $\dot{q}$ | 24:36 |

Table S6: Reward weights

| Symbol | $w_{v_{xy}}$ | $w_{\omega_z}$ | $w_{v_z}$ | $w_{\omega_{xy}}$ | $w_{q_\tau}$ |
|--------|--------------|----------------|-----------|-------------------|-------------|
| Value | 1.0 | 0.5 | $-2.0$ | $-0.05$ | $-2.5e^{-5}$ |

| Symbol | $w_{\ddot{q}}$ | $w_{\dot{a}}$ | $w_{f_a}$ | $w_c$ | $w_g$ |
|--------|----------------|---------------|-----------|-------|-------|
| Value | $-2.5e^{-7}$ | $-0.01$ | 0.5 | $-1.0$ | $-5.0$ |

where $\sigma_{\omega_z} = 0.25$ denotes a temperature factor, $c_z$ and $\omega_z$ denote the commanded and current base angular velocity.

Linear velocity $z$

$$r_{v_z} = w_{v_z} \|v_z\|_2^2 ,$$

where $v_z$ denotes the base vertical velocity.

Angular velocity $x, y$

$$r_{\omega_{xy}} = w_{\omega_{xy}} \|\omega_{xy}\|_2^2 ,$$

where $\omega_{xy}$ denotes the current base roll and pitch velocity.

Joint torque

$$r_{q_\tau} = w_{q_\tau} \|\tau\|_2^2 ,$$

where $\tau$ denotes the joint torques.

Joint acceleration

$$r_{\ddot{q}} = w_{\ddot{q}} \|\ddot{q}\|_2^2 ,$$

where $\ddot{q}$ denotes the joint acceleration.

Action rate

$$r_{\dot{a}} = w_{\dot{a}} \|a' - a\|_2^2,$$

where $a'$ and $a$ denote the previous and current actions.

Feet air time

$$r_{f_a} = w_{f_a} t_{f_a},$$

where $t_{f_a}$ denotes the sum of the time for which the feet are in the air.

Undesired contacts

$$r_c = w_c c_u,$$

where $c_u$ denotes the counts of the undesired knee contacts.

Flat orientation

$$r_g = w_g g_{xy}^2,$$

where $g_{xy}$ denotes the $xy$-components of the projected gravity.

## A.2  NETWORK ARCHITECTURE

### A.2.1  RWM-O

The robotic world model consists of a GRU base and MLP heads predicting the mean and standard deviation of the next observation and privileged information such as contacts, as detailed in Table S7.

Table S7: RWM-O architecture

| Component | Type | Hidden Shape | Activation |
|---|---|---|---|
| base | GRU | 256, 256 | – |
| heads | MLP | 128 | ReLU |

### A.2.2  MOPO-PPO

The network architectures of the policy and the value function used in MOPO-PPO are detailed in Table S8.

Table S8: Policy and value function architecture

| Network | Type | Hidden Shape | Activation |
|---|---|---|---|
| policy | MLP | 128, 128, 128 | ELU |
| value function | MLP | 128, 128, 128 | ELU |

### A.2.3  BASELINES

All MBRL baselines use the **same** architecture and hyperparameters for consistency and fair comparison as detailed in Table S9.

Table S9: Baseline architecture

| Method | MBPO | MBPO-PPO | MOPO | MOPO-PPO |
|---|---|---|---|---|
| Model Architecture | RWM | RWM | RWM-O | RWM-O |
| Policy Optimization | SAC | PPO | SAC | PPO |

## A.3  TRAINING PARAMETERS

The learning networks and algorithm are implemented in PyTorch 2.4.0 with CUDA 12.6 and trained on an NVIDIA RTX 4090 GPU.

### A.3.1  RWM-O

The training information of RWM-O is summarized in Table S10.

Table S10: RWM-O training parameters

| Parameter | Symbol | Value |
|---|---|---|
| step time seconds | $\Delta t$ | 0.02 |
| max iterations | — | 2500 |
| learning rate | — | $1e^{-4}$ |
| weight decay | — | $1e^{-5}$ |
| batch size | — | 1024 |
| ensemble size | $B$ | 5 |
| history horizon | $M$ | 32 |
| forecast horizon | $N$ | 8 |
| forecast decay | $\alpha$ | 1.0 |
| transitions in training data | — | $6M$ |
| approximate training hours | — | 1 |
| number of seeds | — | 5 |

### A.3.2 MOPO-PPO

The training information of MOPO-PPO is summarized in Table S11.

Table S11: MOPO-PPO training parameters

| Parameter | Symbol | Value |
|---|---|---|
| imagination environments | — | 4096 |
| imagination steps per iteration | — | 100 |
| uncertainty penalty weight | $\lambda$ | 1.0 |
| step time seconds | $\Delta t$ | 0.02 |
| buffer size | $|\mathcal{D}|$ | 1000 |
| max iterations | — | 2500 |
| learning rate | — | 0.001 |
| weight decay | — | 0.0 |
| learning epochs | — | 5 |
| mini-batches | — | 4 |
| KL divergence target | — | 0.01 |
| discount factor | $\gamma$ | 0.99 |
| clip range | $\epsilon$ | 0.2 |
| entropy coefficient | — | 0.005 |
| number of seeds | — | 5 |

### A.4 REAL-WORLD DATA COLLECTION

Although RWM-O can be trained on offline data from arbitrary sources, we develop an autonomous pipeline for collecting additional real-world data with minimal human intervention. The robot executes a velocity-tracking policy and records transitions as described in Sec. A.1.1. A visualization of the collection setup is shown in Fig. S7. To ensure safety, a rectangular boundary is defined relative to the odometry origin. Robot odometry is computed by fusing legged odometry and LiDAR data. At each sampling interval, a base velocity command (green arrow) is randomly drawn, and the resulting final pose (red sphere) is predicted via finite integration. If this predicted pose lies outside the safety boundary, a new command is sampled. After a fixed number of retries, a fallback command that drives the robot toward the origin is used instead. To mitigate odometry drift, the safety region can be re-centered around the robot's current position using a joystick override. In total, we collect approximately one hour of data corresponding to 200K state transitions.

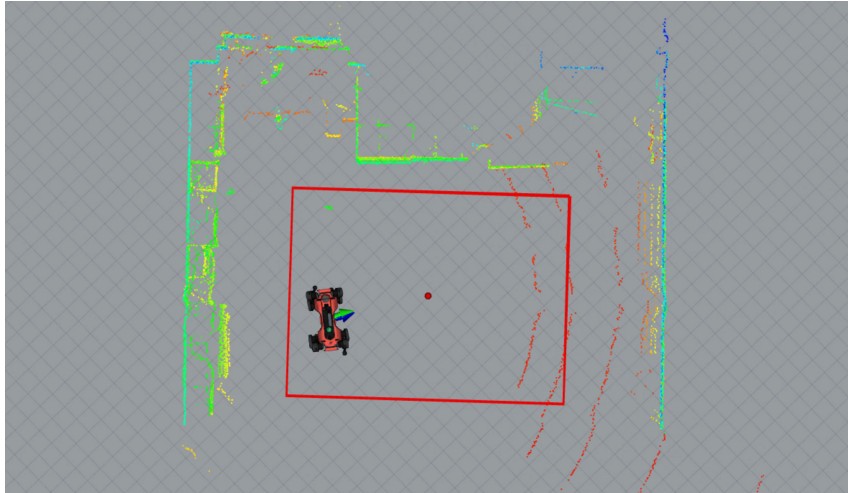

Figure S7: RViz visualization of the real-world data collection process. The green arrow indicates a sampled base velocity command; the red sphere marks the estimated final pose via finite integration. The red box illustrates the safety boundary around the odometry origin, which the robot must remain within during data collection.

## A.5 DISTINCT CHALLENGES

A large body of offline RL research is built upon off-policy, value-function-based methods such as CQL (Kumar et al., 2020), IQL (Kostrikov et al., 2021), and COMBO (Yu et al., 2021). While these approaches achieve strong results on standard offline RL benchmarks, they do not translate directly to high-dimensional real-world locomotion. In contemporary legged robotics, there is substantial empirical evidence that on-policy policy-gradient methods, particularly PPO (Schulman et al., 2017), provide significantly more robust and reliable performance. Although the precise reasons for this difference remain an active research topic, several observations have emerged from practice. Modern locomotion systems rely on massively parallel simulation, where wall-clock training speed rather than sample efficiency dominates the design. PPO's simple update structure and stable optimization behavior make it effective in this regime, whereas off-policy critics such as SAC (Haarnoja et al., 2018) must estimate and update complex value networks that are more difficult to stabilize when scaled to thousands of environments. Off-policy value-based methods also tend to struggle with the distributional shift characteristic of locomotion control. Real-world proprioceptive trajectories are highly correlated, and critic-based objectives are prone to overfitting narrow regions of the state–action space induced by the most recent data distribution. This can lead to brittle policies that generalize poorly and do not transfer reliably to hardware. Existing investigations further highlight the difficulty of tuning SAC-based methods to match PPO's stability in large-scale locomotion training (Raffin, 2025). Despite ongoing research, there is currently no robust recipe for deploying SAC-style approaches in this domain.

These considerations directly constrain the design of offline RL methods capable of operating on physical robots. Offline algorithms built on off-policy value functions cannot be readily applied to locomotion because the resulting policies are not reliably deployable. Moreover, many value-based offline RL approaches are not model-based and therefore cannot leverage synthetic rollouts—an essential capability when no further real-world data can be collected. For locomotion tasks involving long-horizon credit assignment and limited dataset coverage, the ability to generate additional imagined trajectories through a learned dynamics model is crucial. Consequently, this work focuses on model-based and on-policy formulations, which better align with the requirements of fully offline learning in real high-dimensional robotic systems.

Within this context, RWM-O addresses several distinctive challenges inherent to fully offline model-based reinforcement learning. Prior offline MBRL approaches, such as MOPO (Yu et al., 2020), have largely been evaluated in simplified simulated settings and typically employ single-step feedforward ensembles combined with short-horizon value-based optimization. In contrast, our method builds

on the autoregressive world model introduced in RWM, enabling long-horizon dynamics learning suitable for locomotion control. A central aspect of our approach is the ability to quantify and propagate epistemic uncertainty over extended model-generated trajectories. Ensemble disagreement naturally reflects compounding prediction error in regions with limited data support, as illustrated in Fig. 2 (right), and provides a principled mechanism for avoiding unreliable parts of the state–action space.

We further develop MOPO-PPO, which adapts MOPO's uncertainty-penalized training framework to on-policy PPO. This adaptation is *nontrivial*: unlike MOPO's short one-step rollouts with SAC, our system must incorporate uncertainty-aware penalties over autoregressive rollouts spanning up to 100 steps. Maintaining stability in the presence of compounding model uncertainty under a strict fully offline constraint introduces algorithmic challenges distinct from those faced in online or partially interactive settings. Finally, we demonstrate that this uncertainty-aware long-horizon offline MBRL pipeline can be deployed on a physical quadruped, where policies are trained entirely offline from a fixed dataset. To our knowledge, this represents the first demonstration of uncertainty-penalized long-horizon offline MBRL operating reliably on real robotic hardware.

### A.6 ONLINE VS. OFFLINE REINFORCEMENT LEARNING

Reinforcement learning can be conducted under several interaction regimes, and the distinction between online and offline settings has important implications for both algorithm design and achievable performance. In online RL, the learning system is allowed to interact with the environment throughout training. After each policy or model update, the controller can return to the agent to collect additional trajectories, which serve as fresh supervision. This iterative feedback loop enables the system to continuously correct model errors, reduce domain mismatch between predicted and real dynamics, and expand the coverage of the experienced state–action distribution as the policy evolves. Recent robotics works that demonstrate strong real-world learning performance operate within this paradigm (Yang et al., 2020; Smith et al., 2022; Levy et al., 2024). Although some training phases in these works occur offline, each method continually acquires new on-robot data, allowing the model and policy to adapt whenever they become unreliable.

In contrast, the fully offline RL setting considered in this work assumes that no additional environment interaction is permitted once the dataset is collected (Levine et al., 2020). All components of the system—dynamics modeling, long-horizon rollouts, and policy optimization—must operate exclusively on a fixed dataset. This imposes fundamentally different constraints. The dataset may contain only a limited number of behaviors, be biased toward safe or suboptimal trajectories, and lack critical transitions such as failures or recovery maneuvers that are naturally observed during online exploration. Without the ability to gather new samples, the learning system cannot correct model inaccuracies or distributional biases by expanding its coverage of the state–action space. As a result, distribution shift is substantially amplified, model prediction errors accumulate over long autoregressive rollouts, and the policy lacks any external feedback when entering underrepresented regions of the data (Levine et al., 2020; Prudencio et al., 2023).

These differences lead to distinct algorithmic requirements. Whereas online RL can rely on continual data collection to stabilize training, offline RL must depend solely on mechanisms internal to the model and policy optimization process. In particular, uncertainty estimation becomes essential for detecting underexplored states and preventing the policy from drifting into regions where the model is unreliable. The uncertainty-aware design of RWM-O and MOPO-PPO is therefore motivated directly by the challenges inherent to fully offline settings rather than by those faced in online or iterative data-collection regimes.

The contrast between online and offline RL is thus not a minor technical distinction but a defining aspect of the problem formulation. Online methods leverage real-world feedback to mitigate compounding errors and bridge domain gaps, whereas fully offline methods must succeed without such corrective interaction. This work focuses on the latter regime and investigates the design principles required to make fully offline model-based reinforcement learning viable for real robotic systems.

