# OpenReview forum: "Making Offline Model-Based Reinforcement Learning Work on Real Robots"
_ICLR.cc/2026/Conference — Submitted to ICLR 2026_

### Official Review · Reviewer_iN9s · 2025-10-29

**Soundness:** 3
**Presentation:** 4
**Contribution:** 2
**Rating:** 4
**Confidence:** 5

**Summary:**

This paper proposes an offline model-based RL pipeline that can be successfully deployed on real robots. The pipeline consists of two stages. First, they train an autoregressive dynamics model on the offline dataset to predict long-horizon rollouts. The dynamics model captures both aleatoric uncertainty (via Gaussian variance) and epistemic uncertainty (via bootstrap ensembling). Second, they train a policy using an uncertainty-aware model-based PPO algorithm (MOPO-PPO). Specifically, they train the policy on model-imagined rollouts, with the epistemic uncertainty (ensemble variance) subtracted from the reward to encourage conservative behavior. This pipeline demonstrates strong performance gain over uncertainty-unaware baselines across three simulated environments (Reach-Franka, Velocity-G1, Velocity-ANYmal-D), and successfully transfers to a real-world Velocity-ANYmal-D task. Through careful ablation, the paper analyzes the effect of the uncertainty penalty and establishes a correlation between the model's prediction error and uncertainty estimate. Overall, the paper makes a solid empirical contribution towards deploying offline model-based RL in the real world.

**Strengths:**

1. The paper makes several empirical contributions to improve upon prior offline model-based RL methods and to eventually deploy offline model-based RL on a real robot.
2. The paper conducts thorough ablation studies to justify each design choice. They establish a correlation between epistemic uncertainty and model prediction error, justifying the need for the uncertainty penalty. They further analyze the effect of different lambda coefficients for the uncertainty penalty. When lambda is too large, the policy becomes too conservative and stays still. When lambda is too small, the policy starts exploiting the model. Last but not least, they compare different data mixtures, providing valuable insights into what data mixture should be used for offline model-based RL.

**Weaknesses:**

1. Despite the empirical contributions, the paper lacks novelty since every component has been proposed by / studied in prior work (e.g., uncertainty-aware dynamics models, uncertainty-penalized policy optimization).
2. The real-world experiments lack comparison to immediately relevant baselines beyond a hard-coded data collection policy and an online model-free policy.
3. All experiments run from low-dimensional observations. It is unclear how scalable the method is to high-dimensional observations. For one thing, it is hard to quantify epistemic uncertainty using pixel reconstruction.

**Questions:**

**Major**

1. Can you compare to an offline model-based RL baseline in the real-world experiments? This is important since you claimed that your method is "the first demonstration of offline MBRL operating reliably on real robotic hardware." The implication is that other offline MBRL methods fail to do so.
2. A close follow-up to MOPO, COMBO [1], is omitted from all the discussions. Their main argument is to directly learn a conservative Q function instead of using an uncertainty reward penalty. And it seems to work better than MOPO. Can you either justify the omission or provide a comparison, at least in the simulated benchmark?
3. In Table 1, what exactly is the online model-free policy? It seems that on most data mixtures, your method is worse than the online model-free policy. Is this to be expected?

**Minor**

4. Despite the justifications in Appendix A.5, it's still unclear to me why PPO is better than SAC. Can you explain a bit more why this is the case?
5. When rolling out the model, you predict the next observation conditioned on the ensemble mean. Would this potentially lead to mode averaging when the dynamics is multimodal (high aleatoric uncertainty)?
6. Can you add a detailed description of each simulated environment (Reach-Franka, Velocity-G1, Velocity-ANYmal-D)? Currently it is unclear what each task involves.
8. Line 216, "By incorporating uncertainty-aware modeling into the dynamics learning process, RWM-O enables robust trajectory forecasting in offline settings with uncertainty effectively propagated over long horizons." This sentence is confusing.

References:

[1] Tianhe Yu, Aviral Kumar, Rafael Rafailov, Aravind Rajeswaran, Sergey Levine, Chelsea Finn. COMBO: Conservative Offline Model-Based Policy Optimization. NeurIPS 2021.

---

> ### Author Response · Authors · 2025-11-20
>
> Thank you for your time reviewing our work and your valuable feedback. We have improved our paper based on your concerns, as addressed in the following. Please also check the general response, where we updated the paper with the improvements and presented materials.
>
> **Weakness 1**
>
> > Despite the empirical contributions, the paper lacks novelty since every component has been proposed by / studied in prior work (e.g., uncertainty-aware dynamics models, uncertainty-penalized policy optimization).
>
> We thank the reviewer for raising this thoughtful concern. While we agree that the individual components of our system build on established ideas, we believe that the **effective integration** of these components into a **robust, uncertainty-aware, long-horizon offline MBRL pipeline** applicable to real robots represents a meaningful contribution. Our goal is not to introduce algorithmic novelty for its own sake, but to address an underexplored and practically important challenge: **how to make offline MBRL function safely and consistently on real hardware**, where long-horizon predictive accuracy and uncertainty calibration become critical.
>
> As highlighted in **Sec. A.5**, several gaps remain before existing uncertainty-aware MBRL methods can be applied to real systems: substantially amplified distribution shift, compounding autoregressive model errors over hundreds of steps, and limited guidance on how epistemic and aleatoric uncertainty should be structured and penalized in long-horizon rollouts. These issues are central in real robotics but are not addressed in prior work, which largely focuses on short-horizon or highly controlled simulation settings.
>
> This framing aligns with many impactful real-robot learning papers. For instance, [2] explicitly acknowledges that *“our algorithmic framework is not novel… [but] our result is enabled by careful implementation and task setup.”* Similarly, [3] focuses on constructing a principled recipe for real-world deployment rather than introducing new algorithms. These works are well-regarded precisely because they identify the design decisions that make learning succeed on hardware.
>
> Our contribution follows this same philosophy. Rather than proposing a new theoretical building block, we systematically identify and analyze the decisions—uncertainty penalization, autoregressive long-horizon rollout structure, epistemic/aleatoric decomposition, and stable policy learning—that allow offline MBRL to operate reliably under a strict no-interaction constraint on a high-dimensional robot. To our knowledge, this combination has not previously been demonstrated on hardware, and we believe it fills an important gap between offline RL theory and practical robotics. We would highly appreciate the reviewer's recognition and acknowledgment of this line of work.
>
> We appreciate the reviewer’s feedback and made the methodological contribution clearer in the paper.
>
> [2] Smith, L., Kostrikov, I. and Levine, S., 2022. A walk in the park: Learning to walk in 20 minutes with model-free reinforcement learning. arXiv preprint arXiv:2208.07860.
>
> [3] Levy, J., Westenbroek, T. and Fridovich-Keil, D., 2024. Learning to walk from three minutes of real-world data with semi-structured dynamics models. arXiv preprint arXiv:2410.09163.

---

> > ### Author Response · Authors · 2025-11-20
> >
> > **Weakness 2**
> >
> > > The real-world experiments lack comparison to immediately relevant baselines beyond a hard-coded data collection policy and an online model-free policy.
> >
> >
> > We thank the reviewer for raising this important point. In principle, many conservative offline RL methods—such as COMBO [1], CQL [4], and IQL [5]—are relevant baselines. However, these methods all rely on **off-policy value function learning**, a class of algorithms that is known to perform poorly for real-world legged locomotion. In contrast, **on-policy policy gradient methods (e.g., PPO)** remain the de-facto standard for high-dimensional locomotion control. This empirical gap is well documented and is the subject of dedicated investigations (e.g., the analysis in [6]). As a result, value-function-based offline RL methods are **not directly translatable** to our setting, where the policy must ultimately run on a physical quadruped and must be trainable using on-policy policy gradient machinery. We provide a detailed discussion on the challenges we face given such constraints in our updated **Sec. A.5**.
> >
> > This also relates to a second and equally important point: our work is explicitly scoped to **model-based** setups. Offline MBRL is attractive here because imagination rollouts can generate additional synthetic data for policy improvement—an essential capability under the strict fully offline constraint. Many popular offline RL methods, including CQL, IQL, and other critic-based approaches, **do not leverage imagination** and thus do not benefit from the additional training signal that model-based methods uniquely enable in this setting. For locomotion, where long-horizon credit assignment and diverse data coverage are critical, this synthetic-data capability is not optional—it is a necessary assumption of the problem we aim to study.
> >
> > This is precisely one of the motivations of our paper: we aim to highlight that methods performing strongly on simulation benchmarks **do not automatically extend** to real-world robotic applications, where constraints on dynamics, stability, and rollout horizon can make certain algorithmic families unsuitable. For locomotion in particular, these limitations are pronounced: off-policy critics tend to overfit narrow regions of the state-action space induced by large sample sizes from massively parallel training, generalize poorly outside the data distribution predominant with the latest policy samples, and yield policies that do not transfer to hardware.
> >
> > Given these constraints, we focused on baselines **compatible with on-policy optimization**, where uncertainty-aware rollouts can be integrated into the update. **MOPO-PPO** serves precisely this purpose: it adapts the general MOPO framework to policy-gradient-based control and is therefore one of the few uncertainty-aware baselines that can actually be deployed in our locomotion setting.
> >
> > We acknowledge, however, the importance of evaluating whether classic offline RL baselines can perform well in simulation, even if they do not translate to hardware. For this reason, we did include CQL in our simulation study (**L362, Fig. 4**). As shown, while CQL performs reasonably well on manipulation tasks such as Reach-Franka, its performance degrades significantly on locomotion tasks (*Velocity-G1*, *Velocity-Anymal-D*), consistent with the limitations of off-policy value-based methods discussed above. Such policies also proved unsuitable for real-robot deployment. With the reviewer’s suggestion, we also added evaluation of COMBO on our simulated benchmarks, which reveal comparable conclusions as CQL.
> >
> > Overall, the lack of additional real-world baselines is not due to omission, but reflects a **genuine methodological gap**: the space of offline RL algorithms that are both (1) uncertainty-aware and (2) compatible with long-horizon, high-dimensional on-policy policy gradient is currently very limited. Our work aims to make this limitation explicit and provide a step toward bridging this gap.
> >
> > [1] Yu, T., Kumar, A., Rafailov, R., Rajeswaran, A., Levine, S. and Finn, C., 2021. Combo: Conservative offline model-based policy optimization. Advances in neural information processing systems, 34, pp.28954-28967.
> >
> > [4] Kumar, A., Zhou, A., Tucker, G. and Levine, S., 2020. Conservative q-learning for offline reinforcement learning. Advances in neural information processing systems, 33, pp.1179-1191.
> >
> > [5] Kostrikov, I., Nair, A. and Levine, S., 2021. Offline reinforcement learning with implicit q-learning. arXiv preprint arXiv:2110.06169.
> >
> > [6] Raffin, A., Getting SAC to Work on a Massive Parallel Simulator: An RL Journey With Off-Policy Algorithms. https://araffin.github.io/post/sac-massive-sim/

---

> > > ### Author Response · Authors · 2025-11-20
> > >
> > > **Weakness 3**
> > >
> > > > All experiments run from low-dimensional observations. It is unclear how scalable the method is to high-dimensional observations. For one thing, it is hard to quantify epistemic uncertainty using pixel reconstruction.
> > >
> > > We appreciate the reviewer’s question and are glad to clarify this point. While pixel-based dynamics modeling does introduce challenges—particularly higher **aleatoric** uncertainty due to the complexity of predicting high-dimensional observations—we respectfully argue that **epistemic uncertainty quantification itself fundamentally does not depend on the quality of pixel-level prediction**.
> > >
> > > As established in prior work such as PETS [7], epistemic uncertainty reflects **model uncertainty arising from limited or biased data**, and can be estimated from the **disagreement across ensemble members**, regardless of the absolute prediction error. Even for difficult high-dimensional prediction tasks, the variation between ensemble means remains a valid signal of epistemic uncertainty. This is because ensemble disagreement captures the *model’s confidence about the data distribution*, not the raw reconstruction fidelity. Methods such as Random Network Distillation (RND) [8] demonstrate this principle clearly: epistemic uncertainty can be estimated even **without** an explicit prediction target, further supporting that epistemic estimation does not rely on low-dimensional or easily predictable outputs.
> > >
> > > In our ensemble-based model, epistemic uncertainty is derived from **disagreement across prediction heads**, while aleatoric uncertainty is captured through each head’s variance. Even if pixel observations introduce high aleatoric noise, the epistemic term remains well-defined and decoupled from per-step prediction difficulty. Thus, while extending our system to vision-based settings would require additional engineering—for example, using latent-space dynamics models—it would not invalidate the epistemic uncertainty mechanism itself.
> > >
> > >
> > > [7] Chua, K., Calandra, R., McAllister, R. and Levine, S., 2018. Deep reinforcement learning in a handful of trials using probabilistic dynamics models. Advances in neural information processing systems, 31.
> > >
> > > [8] Burda, Y., Edwards, H., Storkey, A. and Klimov, O., 2018. Exploration by random network distillation. arXiv preprint arXiv:1810.12894.
> > >
> > > **Question 1**
> > >
> > > > Can you compare to an offline model-based RL baseline in the real-world experiments? This is important since you claimed that your method is "the first demonstration of offline MBRL operating reliably on real robotic hardware." The implication is that other offline MBRL methods fail to do so.
> > >
> > >
> > > We thank the reviewer for the question. As discussed in our response to **Weakness 2**, the space of offline model-based RL methods that are compatible with long-horizon, high-dimensional locomotion and on-policy policy-gradient optimization is extremely limited. Most established offline MBRL algorithms rely on off-policy value-function learning, which is known to perform poorly for locomotion and does not yield deployable hardware policies.
> > >
> > > With the reviewer's suggestions, we additionally evaluated COMBO, together with CQL, which was initially discussed as a strong model-free offline RL baseline. As shown in the updated **Fig. 4**, both COMBO and CQL perform well on manipulation (*Reach-Franka*) but degrade significantly on locomotion tasks (*Velocity-G1*, *Velocity-Anymal-D*). This degradation is once again attributed to the failure mode of value-function-based methods in legged locomotion (Sec. A.5). Because these policies fail even in simulation under the offline constraint, they do not provide a viable candidate for real-world deployment.
> > >
> > > At present, other than the general formulation in MOPO, we are not aware of an existing **offline model-based** RL method that (1) supports long-horizon synthetic rollouts, (2) integrates with on-policy policy gradients required for locomotion, and (3) produces stable behavior suitable for hardware testing. Thus, the absence of a real-world offline MBRL baseline reflects a genuine methodological gap rather than an omission. Our analysis aims to make this gap explicit.

---

> > > > ### Author Response · Authors · 2025-11-20
> > > >
> > > > **Question 2**
> > > >
> > > > > A close follow-up to MOPO, COMBO [1], is omitted from all the discussions. Their main argument is to directly learn a conservative Q function instead of using an uncertainty reward penalty. And it seems to work better than MOPO. Can you either justify the omission or provide a comparison, at least in the simulated benchmark?
> > > >
> > > > We thank the reviewer for bringing up COMBO and agree that it is an important and influential offline MBRL method. As discussed in our response to **Weakness 2** and **Question 1**, our real-world setting requires compatibility with on-policy policy-gradient optimization and long-horizon model-based rollouts, both of which are known to be challenging for off-policy, value-function-based methods. For this reason, COMBO, like CQL and IQL, is not directly applicable in our hardware experiments.
> > > >
> > > > We appreciate the reviewer’s suggestion and have conducted additional simulation evaluations to examine how COMBO performs in our benchmark tasks as mentioned above. In fact, COMBO performs comparably to CQL as a model-free off-policy value-function-based pipeline, but again fails to translate to locomotion tasks. We have updated **Section 5.3** and **Fig. 4** to reflect this new comparison.
> > > >
> > > >
> > > > **Question 3**
> > > >
> > > > > In Table 1, what exactly is the online model-free policy? It seems that on most data mixtures, your method is worse than the online model-free policy. Is this to be expected?
> > > >
> > > > We appreciate the reviewer’s question and are happy to clarify. The **online model-free policy** in **Table 1** corresponds to a **well-tuned PPO controller trained entirely in a high-fidelity simulator**, following the standard sim-to-real pipeline widely used in contemporary legged locomotion research. This represents the strongest simulation-based baseline available for our hardware platform.
> > > >
> > > > The purpose of this comparison is to illustrate that our offline MBRL approach, when provided with an appropriate mixture of simulation and real data, can **outperform a state-of-the-art simulator-trained PPO policy**—a capability that pure sim-to-real training does not have, since it cannot incorporate real-world data directly.
> > > >
> > > > Regarding the reviewer’s observation: yes, it is expected that our performance degrades when the dataset contains **excessive amounts of real data**. As discussed in **L421–424**, real-world datasets lack coverage of risky or failure states (e.g., collisions, falls), because collecting such data on hardware is unsafe or impractical. When the real-data portion dominates the mixture, the learned world model becomes **biased toward “good” trajectories** and underestimates underexplored but critical regions. This leads to poor imagination rollouts and consequently weaker offline policy optimization.
> > > >
> > > > This behavior is a well-known limitation of offline RL: the achievable performance is constrained by the **breadth and coverage** of the offline dataset. Our results reflect this expected trade-off.

---

> > > > > ### Author Response · Authors · 2025-11-20
> > > > >
> > > > > **Question 4, Minor**
> > > > >
> > > > > > Despite the justifications in Appendix A.5, it's still unclear to me why PPO is better than SAC. Can you explain a bit more why this is the case?
> > > > >
> > > > > We appreciate the reviewer’s request for clarification. As also discussed in **Weakness 2**, there is substantial empirical evidence that **off-policy value-function-based methods such as SAC** tend to perform poorly in real-world legged locomotion, while **on-policy policy-gradient methods such as PPO** consistently produce state-of-the-art results. Although the precise reasons for this gap are still an active area of investigation, several hypotheses have been put forward in the literature and by practitioners:
> > > > >
> > > > > 1. **Massively parallel simulation favors PPO-like updates.**
> > > > >
> > > > > Contemporary locomotion training pipelines rely on large-scale parallel environments where **wall-clock training speed**, not sample efficiency, is the main bottleneck. PPO is well-suited for this regime due to its simple optimization structure and stable minibatch updates. In contrast, SAC and related critic-based methods require more complex target networks and value estimates, making them more sensitive to instability when scaled to thousands of parallel workers.
> > > > >
> > > > >
> > > > > 2. **Off-policy critics struggle with distributional shift in locomotion.**
> > > > >
> > > > > Off-policy value-based methods are prone to overfitting narrow regions of the state–action space when trained with highly correlated proprioceptive data. This can lead to brittle policies that fail to generalize beyond the most recent distribution of rollouts. PPO, being on-policy, avoids this accumulation of off-policy error.
> > > > >
> > > > >
> > > > > 3. **Existing studies report persistent difficulty tuning SAC for locomotion.**
> > > > >
> > > > > As documented in analyses such as [6], achieving competitive locomotion performance with SAC in massively parallel settings requires substantial nontrivial reengineering, and even then tends to remain less stable than PPO. To date, there is no established recipe for making SAC perform comparably to PPO in these tasks.
> > > > >
> > > > >
> > > > > Taken together, the field has converged—largely through empirical experience—on PPO as the **most robust and reliable baseline for locomotion**, both in sim-to-real pipelines and in real-world deployments. This practical constraint also shapes our work: because our goal is to study offline MBRL for real robots, we adopt the policy-gradient family that is currently known to transfer reliably to hardware.
> > > > >
> > > > > We agree that understanding the deeper causes of the PPO–SAC gap is an important research direction, but at present, the empirical reality is that **SAC-style methods do not provide a stable foundation for legged locomotion**, especially under the additional constraints of offline MBRL. We appreciate that the reviewer raised this concern and thus included a discussion on value-function-based methods in **Sec. A.5**.
> > > > >
> > > > > [6] Raffin, A., Getting SAC to Work on a Massive Parallel Simulator: An RL Journey With Off-Policy Algorithms. https://araffin.github.io/post/sac-massive-sim/
> > > > >
> > > > >
> > > > > **Question 5, Minor**
> > > > >
> > > > > > When rolling out the model, you predict the next observation conditioned on the ensemble mean. Would this potentially lead to mode averaging when the dynamics is multimodal (high aleatoric uncertainty)?
> > > > >
> > > > > We thank the reviewer for this insightful question. In our specific setting—**low-level joint-space locomotion control**—the underlying robot dynamics are largely **deterministic and single-modal**, so multimodality is not a major factor in practice. Nonetheless, we agree that in domains with genuinely multimodal dynamics, mode averaging is an important consideration.
> > > > >
> > > > > As the reviewer notes, conditioning rollouts on the **ensemble mean** may introduce averaging effects. However, we would like to emphasize that **mode averaging can occur even before the averaging step**, because each ensemble member is trained independently on the full dataset. If the true dynamics contain multiple distinct modes but the model class is unimodal, each ensemble member may already collapse to a smoothed approximation of the modes during training. In such cases, avoiding mode averaging would require **different model architectures**, such as mixture-of-experts, gated networks, or diffusion-style generative models, that are explicitly designed to preserve multimodality.
> > > > >
> > > > > While exploring these architectures is beyond the scope of the present work, we expect that the **uncertainty-penalized policy optimization principle** used here would remain applicable: regardless of the specific model class, the policy should avoid regions where the model exhibits high epistemic uncertainty. Incorporating multimodal dynamics models is a promising future direction, and we appreciate the reviewer for highlighting this aspect.

---

> > > > > > ### Author Response · Authors · 2025-11-20
> > > > > >
> > > > > > **Question 6, Minor**
> > > > > >
> > > > > > > Can you add a detailed description of each simulated environment (Reach-Franka, Velocity-G1, Velocity-ANYmal-D)? Currently it is unclear what each task involves.
> > > > > >
> > > > > >
> > > > > > We thank the reviewer for pointing this out. These environments are **standard benchmark tasks provided in IsaacLab [7]**, and we now clarify their definitions more explicitly.
> > > > > >
> > > > > > As shown in **Fig. 5**,
> > > > > > - *Reach-Franka* is a 7-DoF manipulation task in which the Franka Panda arm tracks a target end-effector pose.
> > > > > > - *Velocity-G1* and *Velocity-ANYmal-D* are locomotion tasks on the Unitree G1 humanoid and ANYbotics ANYmal-D quadruped, respectively, where the controller must follow commanded base linear and angular velocities.
> > > > > >
> > > > > >
> > > > > > For completeness, the revised manuscript now includes detailed descriptions of all environments. In addition, we included videos of these results on our [webpage](https://sites.google.com/view/iclr2026-rwm-o/home). Full task details are also documented in the [IsaacLab environment reference](https://isaac-sim.github.io/IsaacLab/main/source/overview/environments.html)
> > > > > >
> > > > > > [7] Mittal, M., Yu, C., Yu, Q., Liu, J., Rudin, N., Hoeller, D., Yuan, J. L., Singh, R., Guo, Y., Mazhar, H., & Mandlekar, A. (2023). Orbit: A unified simulation framework for interactive robot learning environments. IEEE Robotics and Automation Letters, 8(6), 3740–3747.
> > > > > >
> > > > > > **Question 7, Minor**
> > > > > >
> > > > > > > Line 216, "By incorporating uncertainty-aware modeling into the dynamics learning process, RWM-O enables robust trajectory forecasting in offline settings with uncertainty effectively propagated over long horizons." This sentence is confusing.
> > > > > >
> > > > > > We thank the reviewer for pointing out the ambiguity. The intended meaning is the following:
> > > > > > As illustrated in **Fig. 2 (right)**, the **epistemic uncertainty** produced by the ensemble reflects the model’s **autoregressive prediction error over long horizons**. During inference, at each step of the rollout, the model outputs both the predicted next state and its corresponding epistemic uncertainty. As the autoregressive trajectory progresses, errors in regions with limited or biased data naturally accumulate. When the predicted trajectory begins to drift outside well-supported parts of the state–action space, this deviation is captured by a **progressive increase in the epistemic uncertainty estimate**.
> > > > > >
> > > > > > We have revised the sentence in the manuscript to convey this more clearly.

---

### Official Review · Reviewer_oixy · 2025-10-29

**Soundness:** 1
**Presentation:** 4
**Contribution:** 2
**Rating:** 2
**Confidence:** 3

**Summary:**

The contribution proposes an offline MBRL algorithm and evaluates it in simulation and on real hardware. The MBRL algorithm relies on an autoregressive robot world model. Ensembles are used to capture epistemic and aleatoric model uncertainty. The policy optimization penalizes exploration in regions with high epistemic uncertainty. This way overfitting to dynamics model errors is discouraged. The method is evaluated in simulation and on a real-world hardware experiment.

**Strengths:**

The paper is very well written and results are presented nicely. The proposed approach seems reasonable and is well explained. The hardware application is impressive and comparisons in simulations are exhaustive.

**Weaknesses:**

- The central and strong claim of this work represented in the abstract and title is that this is the '[...] first demonstration of offline MBRL operating reliably on real robotic hardware.' If I understand correctly, at least [1] (See Sec. IV - algorithms) and maybe to some extend [2] evaluated offline model-based reinforcement learning on real robotic hardware. Therefore, one cannot claim that this work is the first application of offline MBRL on real hardware. What "reliable" means in the claim is not further defined. With this very strong claim I would have at least expected an exhaustive review on related hardware applications of offline MBRL and why the application presented in this paper is more significant and indeed the first reliable one. The claim of the title and the abstract is somewhat softened in the introduction: "this is the first demonstration of uncertainty-penalized offline MBRL operating reliably on a physical robot". If this is the real claim then the abstract and title should be changed accordingly in my opinion.

- The authors mention in Sec. 2.2 that there are many methods to incorporate uncertainty estimation into MBRL ([3] may be an additional relevant related work here as they achieve a rollout length of around 30 steps on average). The paper differentiates itself from those methods solely by mentioning that those methods have not been applied on hardware "While these methods achieve impressive performance in controlled simulation benchmarks, applying them to real-world robotics remains a significant hurdle, where reliability and robustness demand both accurate long-horizon modeling and stable policy learning." This makes it hard to evaluate what the methodological contribution of the paper is. Additionally, it seems that none of the baselines chosen in Sec. 5 were uncertainty aware. Therefore, results support that uncertainty awareness is important in general but do not support the effectiveness of the method proposed in this paper compared to others in the uncertainty aware space.

Additionally, I am unsure if ICLR is the right venue for the paper. Since the main contribution is to "make ... learning work on real robots" I'd suspect a robotics venue might be more fitting.


[1] G. Zhou, L. Ke, S. Srinivasa, A. Gupta, A. Rajeswaran and V. Kumar, "Real World Offline Reinforcement Learning with Realistic Data Source," 2023 IEEE International Conference on Robotics and Automation (ICRA), London, United Kingdom, 2023, pp. 7176-7183, doi: 10.1109/ICRA48891.2023.10161474.

[2] X. Li, W. Shang and S. Cong, "Offline Reinforcement Learning of Robotic Control Using Deep Kinematics and Dynamics," in IEEE/ASME Transactions on Mechatronics, vol. 29, no. 4, pp. 2428-2439, Aug. 2024, doi: 10.1109/TMECH.2023.3336316.

**Questions:**

- I would suggest making the central claim of the paper '[...] first demonstration of offline MBRL operating reliably on real robotic hardware.' more specific and adding a literature review to support the more specific claim.

---

> ### Author Response · Authors · 2025-11-20
>
> Thank you for your time reviewing our work and your valuable feedback. We have improved our paper based on your concerns, as addressed in the following. Please also check the general response, where we updated the paper with the improvements and presented materials.
>
> **Weakness 1**
>
> > The central and strong claim of this work represented in the abstract and title is that this is the '[...] first demonstration of offline MBRL operating reliably on real robotic hardware.' If I understand correctly, at least [1] (See Sec. IV - algorithms) and maybe to some extend [2] evaluated offline model-based reinforcement learning on real robotic hardware. Therefore, one cannot claim that this work is the first application of offline MBRL on real hardware. What "reliable" means in the claim is not further defined. With this very strong claim I would have at least expected an exhaustive review on related hardware applications of offline MBRL and why the application presented in this paper is more significant and indeed the first reliable one. The claim of the title and the abstract is somewhat softened in the introduction: "this is the first demonstration of uncertainty-penalized offline MBRL operating reliably on a physical robot". If this is the real claim then the abstract and title should be changed accordingly in my opinion.
>
> We thank the reviewer for bringing these related works to our attention. While [1] represents a model-free offline RL approach, we appreciate being made aware of [2] and agree that both are relevant to include in our introduction discussion. We also agree that our initial phrasing (“first demonstration of offline MBRL operating reliably on real robotic hardware”) may have been interpreted more broadly than intended. To avoid any possible confusion, we have **removed** this strong claim from the abstract and revised the introduction to describe our contribution more precisely as the **first demonstration of an uncertainty-penalized offline MBRL pipeline operating reliably on a high-dimensional robot**. We hope the distinctive challenges this work attempts to tackle are well explained in our **Section A.5**.
>
> Regarding the reviewer’s question about the meaning of “reliable,” we appreciate the opportunity to clarify this point. In our context, “reliable” refers to the ability of the learned controller to achieve **consistent and strong performance on hardware on full-scale, non-simplified tasks** despite operating purely from a fixed dataset, especially **in the same setup** as the state-of-the-art alternatives (simulator-based model-free policy). As shown in **Table 1**, our offline-trained policy achieves **0.91** in hardware evaluation, compared to **0.88** from a well-trained simulator-to-real PPO baseline. Although this improvement is not visually dramatic, we view it as scientifically meaningful: **improving upon a simulation-trained controller on non-simplified omnidirectional velocity tracking tasks without any new data** highlights the practical effectiveness of the uncertainty-aware offline MBRL pipeline developed in this work.
>
> Thanks to the suggestion, we included the mentioned works, removed the seemingly strong claim, and clarified what we meant by “reliable” in our improved manuscript.

---

> > ### Author Response · Authors · 2025-11-20
> >
> > **Weakness 2**
> >
> > > The authors mention in Sec. 2.2 that there are many methods to incorporate uncertainty estimation into MBRL ([3] may be an additional relevant related work here as they achieve a rollout length of around 30 steps on average). The paper differentiates itself from those methods solely by mentioning that those methods have not been applied on hardware "While these methods achieve impressive performance in controlled simulation benchmarks, applying them to real-world robotics remains a significant hurdle, where reliability and robustness demand both accurate long-horizon modeling and stable policy learning." This makes it hard to evaluate what the methodological contribution of the paper is. Additionally, it seems that none of the baselines chosen in Sec. 5 were uncertainty aware. Therefore, results support that uncertainty awareness is important in general but do not support the effectiveness of the method proposed in this paper compared to others in the uncertainty aware space.
> >
> > We thank the reviewer for raising this thoughtful concern. While we agree that the individual components of our system build on established ideas, we believe that the **effective integration** of these components into a **robust, uncertainty-aware, long-horizon offline MBRL pipeline** applicable to real robots represents a meaningful contribution. Our goal is not to introduce algorithmic novelty for its own sake, but to address an underexplored and practically important challenge: **how to make offline MBRL function safely and consistently on real hardware**, where long-horizon predictive accuracy and uncertainty calibration become critical.
> >
> > As highlighted in **Sec. A.5**, several gaps remain before existing uncertainty-aware MBRL methods can be applied to real systems: substantially amplified distribution shift, compounding autoregressive model errors over hundreds of steps, and limited guidance on how epistemic and aleatoric uncertainty should be structured and penalized in long-horizon rollouts. These issues are central in real robotics but are not addressed in prior work, which largely focuses on short-horizon or highly controlled simulation settings.
> >
> > This framing aligns with many impactful real-robot learning papers. For instance, [4] explicitly acknowledges that *“our algorithmic framework is not novel… [but] our result is enabled by careful implementation and task setup.”* Similarly, [5] focuses on constructing a principled recipe for real-world deployment rather than introducing new algorithms. These works are well-regarded precisely because they identify the design decisions that make learning succeed on hardware.
> >
> > Our contribution follows this same philosophy. Rather than proposing a new theoretical building block, we systematically identify and analyze the decisions—uncertainty penalization, autoregressive long-horizon rollout structure, epistemic/aleatoric decomposition, and stable policy learning—that allow offline MBRL to operate reliably under a strict no-interaction constraint on a high-dimensional robot. To our knowledge, this combination has not previously been demonstrated on hardware, and we believe it fills an important gap between offline RL theory and practical robotics.
> >
> > Regarding the reviewer’s concern about baselines, our intention was not to compare different uncertainty-estimation techniques or to claim superiority of one uncertainty mechanism over another. Rather, our goal was to demonstrate that **incorporating uncertainty at all** is essential for achieving stable long-horizon offline MBRL on real hardware. In this sense, our inclusion of **MOPO-PPO** is meant to validate this principle: MOPO’s core idea—penalizing uncertainty during synthetic rollouts—is representative of common uncertainty-aware MBRL practices, and our adaptation to PPO extends this uncertainty quantification to **long-horizon autoregressive modeling**, which is necessary for deployment on real robotic systems.
> >
> > We appreciate the reviewer’s feedback and made the methodological contribution clearer in the paper.
> >
> > [4] Smith, L., Kostrikov, I. and Levine, S., 2022. A walk in the park: Learning to walk in 20 minutes with model-free reinforcement learning. arXiv preprint arXiv:2208.07860.
> >
> > [5] Levy, J., Westenbroek, T. and Fridovich-Keil, D., 2024. Learning to walk from three minutes of real-world data with semi-structured dynamics models. arXiv preprint arXiv:2410.09163.

---

> > > ### Author Response · Authors · 2025-11-20
> > >
> > > **Suggestion 1**
> > >
> > > > Additionally, I am unsure if ICLR is the right venue for the paper. Since the main contribution is to "make ... learning work on real robots" I'd suspect a robotics venue might be more fitting.
> > >
> > > We appreciate the reviewer’s suggestion and understand the concern. At the same time, we would like to share a broader perspective that motivated us to submit this work to ICLR. In our experience, research that aims to bridge learning theory and real robotic deployment often finds itself in a **difficult middle ground**: within the robotics community, methods that deviate from the dominant “model-free, simulator-first” paradigm can be difficult to position, while within the machine learning community, the substantial effort required to make learning-based frameworks succeed on real hardware is not always visible or fully appreciated.
> > >
> > > Our intention with this submission is to contribute to closing this gap. Offline MBRL is fundamentally a **machine learning problem**—rooted in uncertainty modeling, distribution shift, and long-horizon structure—yet demonstrating its feasibility in the real world requires significant systems insight. We believe ICLR is an appropriate venue because it brings together researchers who care deeply about both methodological advances and their empirical grounding, as so it also notes “applications to robotics” as one of its [subject areas](https://iclr.cc/Conferences/2026/CallForPapers). We hope that presenting a careful, uncertainty-aware offline MBRL pipeline that operates reliably on a high-dimensional robot can help connect these perspectives and encourage further dialogue between the two communities. We would highly appreciate the reviewer's recognition and acknowledgment of this line of work.
> > >
> > > **Question 1**
> > >
> > > > I would suggest making the central claim of the paper '[...] first demonstration of offline MBRL operating reliably on real robotic hardware.' more specific and adding a literature review to support the more specific claim.
> > >
> > > We thank the reviewer for this helpful suggestion. In response, we have removed the broad claim from the abstract and title and revised the introduction to provide a more precise description of our contribution. As detailed in our response to **Weakness 1**, we also expanded the related-work section to include additional prior hardware demonstrations of offline RL. The revised manuscript now states our contribution more narrowly and accurately, and the literature review has been updated to support this refined claim.

---

### Official Review · Reviewer_uffN · 2025-10-31

**Soundness:** 2
**Presentation:** 3
**Contribution:** 2
**Rating:** 4
**Confidence:** 5

**Summary:**

The paper proposes the framework Offline Robotics World Model  (RWM-O), which extends the previous contribution Robotics World Model (RWM) to the offline setting by adding uncertainty regularizations to the predictions of the dynamics model. The paper also introduces MOPO-PPO, which extends MOPO to use PPO as the base policy optimization algorithm. Finally, the framework uses uncertainty-regularized rewards to prevent the learned policy from drifting from the data distribution. The paper compares to standard baselines from the RL literature on simulated benchmarks and also demonstrates the algorithm on a real quadruped robot.

**Strengths:**

$\textbf{Clarity}$: The paper is well written, and it is easy to understand how the many components fit together.

$\textbf{Thorough Evaluations}$: The paper carefully ablates key design decisions and demonstrates how they must be tuned to achieve strong results.

$\textbf{Real world results:}$ The paper demonstrates that the algorithm can be applied directly to a high-dimensional real-world quadruped.

**Weaknesses:**

$\textbf{Missing Related Work}$: The paper largely connects to related work presented at mainline machine learning venues, but misses many developments in applying MBRL (and RL more generally) presented at robotics venues. Given the heavy emphasis the paper places on practical real-world deployment, omitting these works is a major weakness. For example, the following papers apply real-world RL to quadrupeds:

- “A Walk in the Park: Learning to Walk in 20 Minutes With Model-Free Reinforcement Learning” (Smith et al., RSS 2023)

- “Learning to Walk from Three Minutes of Real-World Data with Semi-structured Dynamics Models” (Levy et al., CoRL 2024)

- “Date-Efficient Reinforcement Learning for Legged Robots” (Yang et al., CoRL 2020)

Specifically, the final two papers use uncertainty mitigation techniques to get MBRL to work for real world quadrupeds in a batch offline RL setting (i.e. multiple rounds of collecting data, then updating the policy with offline MBRL). Doing fully offline RL vs. batch offline RL is a minor distinction from prior work, in the opinion of this reviewer.



$\textbf{Real world results are not Surprising}$:  As a concrete examples (Yang et al., CoRL 2020) and (Levy et al., CoRL 2024) learn locomotion policies with only ~40k and ~20k real world environment steps (if my math is correct). This is around an order of magnitude less real world data than the experiments presented in this paper. This is not meant as a direct comparison, as the experimental set ups are obviously different. However, I note this to highlight that I do not find the presented results surprising or impressive, given what has been accomplished previously.

$\textbf{Benchmarks:}$ I question whether the benchmark experiments are informative about what will happen in the real world. Specifically, the current results only demonstrate significant gains when optimal expert data is available. However, I do not think this is a realistic assumption for systems such as quadrupeds — if we had optimal data, wouldn’t we already have an optimal real world policy? What is the real world benefit of the method if it doesn’t show gains when only sub-optimal data is available? }


$\textbf{Minimal Technical Contribution}$: I’m impressed that the authors brought together many different techniques and got them to actually work on a real robot. However, this style of paper only makes a strong publication if the results are surprisingly strong. Given the previous works mentioned above, I do not believe the paper passes this bar in its current form, making the limited technical novelty an additional weak point.

**Questions:**

- Why is offline MBRL the correct approach for real word learning for the tasks in the paper? Given that the current results do not “move the needle” in terms of capabilities, I believe this point needs to be thoroughly defended.

- Can the given approach succeed when optimal data is not available?

---

> ### Author Response · Authors · 2025-11-20
>
> Thank you for your time reviewing our work and your valuable feedback. We have improved our paper based on your concerns, as addressed in the following. Please also check the general response, where we updated the paper with the improvements and presented materials.
>
> **Weakness 1**
>
> > Missing related work: The paper largely connects to related work presented at mainline machine learning venues, but misses many developments in applying MBRL (and RL more generally) presented at robotics venues. Given the heavy emphasis the paper places on practical real-world deployment, omitting these works is a major weakness.
>
> > Specifically, the final two papers use uncertainty mitigation techniques to get MBRL to work for real world quadrupeds in a batch offline RL setting (i.e. multiple rounds of collecting data, then updating the policy with offline MBRL). Doing fully offline RL vs. batch offline RL is a minor distinction from prior work, in the opinion of this reviewer.
>
>
> We thank the reviewer for highlighting these relevant robotics-venue works and agree that they are important to cite given the paper’s emphasis on real-world deployment. We included the suggested works in the revised manuscript.
>
> However, it is important to emphasize that the setup studied in these works is **fundamentally different** from the fully offline setting considered in our paper. All three cited methods fall under **online or batch-online RL**, where additional environment interactions are collected after each training phase. Even in the “batch offline” formulations used in *Levy et al.* and *Yang et al.*, the system repeatedly gathers new on-robot data for the purpose of correcting the learned model and policy. This continual access to the real robot acts as a powerful feedback channel that repeatedly closes the domain gap, simplifies model learning, and prevents compounding model errors from accumulating unchecked.
>
> In contrast, as clarified in L054–056 and L061–065 of the manuscript, our work explicitly addresses the **fully offline RL** setting, where *no further environment interaction is permitted after the dataset is given* [1]. This distinction is not minor: it fundamentally changes both the difficulty and the algorithmic requirements. Without the ability to collect new data, the world model must generalize strictly within the support of a fixed (and often biased) dataset, and the policy must avoid drifting into regions where the model has no corrective signal. A primary motivation for the uncertainty-aware design of RWM-O and MOPO-PPO is precisely this challenge: without additional interaction, epistemic uncertainty becomes the *only* mechanism to detect and avoid underexplored regions.
>
> We further note that although *Levy et al. (CoRL’24)* incorporate epistemic uncertainty via PETS for generating imagined trajectories, their policy update does not explicitly use uncertainty for regularization, and the model is continually updated with new real-world data across batches—again placing it squarely in the online or batch-online regime. Similarly, *Yang et al. (CoRL’20)* assumes iterative access to fresh data, significantly reducing the severity of distribution shift.
>
> Thus, while deployment aspects are indeed similar, the **problem formulation and constraints differ substantially**. Fully offline MBRL removes the primary mechanism existing robotics methods rely on for stabilizing training—online data collection—and therefore tackles a strictly harder problem. We made this distinction clearer and revised the related-work section accordingly. We also included a section in the appendix to explicitly contrast “fully offline RL” with “batch offline/online MBRL” to avoid future ambiguity.
>
> We are thankful that the reviewer revealed a potential misunderstanding of our fundamental setup and therefore faced challenges. To address this, we included a dedicated **Section A.6** to highlight the difference between online and offline RL in our revised manuscript.
>
> [1] Levine, S., Kumar, A., Tucker, G. and Fu, J., 2020. Offline reinforcement learning: Tutorial, review, and perspectives on open problems. arXiv preprint arXiv:2005.01643.

---

> > ### Author Response · Authors · 2025-11-20
> >
> > **Weakness 2**
> >
> > > Real world results are not surprising: As a concrete examples (Yang et al., CoRL 2020) and (Levy et al., CoRL 2024) learn locomotion policies with only ~40k and ~20k real world environment steps (if my math is correct). This is around an order of magnitude less real world data than the experiments presented in this paper. This is not meant as a direct comparison, as the experimental set ups are obviously different. However, I note this to highlight that I do not find the presented results surprising or impressive, given what has been accomplished previously.
> >
> > We appreciate the reviewer’s point but would respectfully highlight that our goal is not to present the strongest real-world locomotion controller to date, but to evaluate—and stress-test—the feasibility of **fully offline MBRL** under realistic robotic conditions, where **no additional data collection is permitted**. As discussed in our response to **Weakness 1**, this setting is fundamentally different from the online or batch-online setups in prior work, where new on-robot data is continuously gathered to correct the model and reduce domain gap. These settings address different research questions, and direct comparisons of sample counts or performance should be made with this distinction in mind. As mentioned before, we kindly invite the reviewer to our added Section A.6 for a detailed discussion on the different challenges between online and offline setups.
> >
> > Only after this conceptual distinction is clear do we note that—even if we compare directly—**the task settings also differ substantially**. The cited methods *(Yang et al., CoRL 2020; Levy et al., CoRL 2024)* focus on forward, fixed-velocity walking, while our experiments tackle **omnidirectional velocity command following**, which demands broader state–action coverage. Hence, the raw number of transitions used in our experiments does not directly reflect lower efficiency, but rather the increased complexity of the control objective.
> >
> > It is more important to recognize that the hardware performance in the cited works is **not** intended to—and indeed does not—reach the capabilities of state-of-the-art model-free controllers trained in high-fidelity simulators (e.g., *Lee et al., Science Robotics 2020* [2]). Their main contribution lies in demonstrating that online or batch-online learning is feasible under limited simulator access. Our contribution parallels this perspective but operates under **strictly more constrained conditions**: unlike these methods, our setting does not allow any new real-world data to be collected during training. This restriction substantially amplifies distribution shift, compounding model errors, and generalization challenges—without the corrective feedback loop that online RL benefits from.
> >
> > Despite these constraints and the higher task complexity, our approach achieves **real-world performance comparable to, and even exceeding, strong simulation-based baselines**. As shown in **Table 1**, a well-trained PPO policy deployed from simulation achieves 0.88 hardware performance, whereas our fully offline pipeline reaches **0.91** using a combination of simulation data and only 200k real transitions. While this improvement is not visually dramatic, it is scientifically meaningful: **surpassing a simulator-trained policy without any opportunity to collect new data is highly non-trivial**. Achieving such performance under the far stricter constraints of offline RL demonstrates that principled uncertainty-aware model design and policy optimization can make offline MBRL viable for real robotics.
> >
> > Finally, as highlighted in **Section A.5**, much of offline MBRL has been validated only in simplified simulated benchmarks. Its behavior on real systems under strict data constraints remains underexplored. Our real-world experiments therefore aim not to set new performance records, but to **systematically evaluate how offline MBRL behaves under real hardware conditions** and to identify the practical design choices that make it work reliably without any additional interaction.
> >
> > [2] Lee, J., Hwangbo, J., Wellhausen, L., Koltun, V., & Hutter, M. (2020). Learning quadrupedal locomotion over challenging terrain. Science Robotics.

---

> > > ### Author Response · Authors · 2025-11-20
> > >
> > > **Weakness 3**
> > >
> > > > Benchmarks: I question whether the benchmark experiments are informative about what will happen in the real world. Specifically, the current results only demonstrate significant gains when optimal expert data is available. However, I do not think this is a realistic assumption for systems such as quadrupeds — if we had optimal data, wouldn’t we already have an optimal real world policy? What is the real world benefit of the method if it doesn’t show gains when only sub-optimal data is available?
> > >
> > > We respectfully point out that the data yielding the highest policy performance in our benchmarks does **not** need to be optimal. While expert-quality data can certainly help—as it biases the model toward dynamics induced by higher-performing behaviors (**Fig. 4; L375–377**)—the core objective of offline RL is to **improve beyond the behavior represented in the dataset**, even when the dataset itself is suboptimal. Our results demonstrate this property clearly in the real-world experiments.
> > >
> > > As described in **L413–417** and **Table 1**, the offline dataset used for real-world training is collected by a **heavily randomized, suboptimal policy** that achieves **only 0.79** on hardware. Despite this, our offline MBRL pipeline attains an average performance of **0.91**, substantially outperforming the behavior policy that created the data (**L417-419**). This illustrates precisely the value of offline MBRL: *it can distill a stronger policy than the one that generated the data*, even when the available data is far from optimal.
> > >
> > > We agree that real-world quadruped systems rarely come with “optimal” data, and our results directly reflect this reality. The dataset we use is intentionally suboptimal and noisy, and the method still achieves strong improvements. More generally, the performance of any offline RL method is necessarily bounded by the quality and coverage of the fixed dataset (**L463-465**). High-quality expert data will raise that ceiling, but the scientific focus is on the **delta** between dataset behavior and learned policy—not the absolute final performance. Our results show that this delta is significant both in simulation and in real-world deployment.
> > >
> > > Thus, the method’s real-world benefit is clear: even with **imperfect, biased, and suboptimal data**—which is the norm in robotics—offline MBRL can still produce policies that outperform the underlying dataset and are suitable for deployment. This directly addresses the reviewer’s concern and aligns with the fundamental goal of offline RL.

---

> > > > ### Author Response · Authors · 2025-11-20
> > > >
> > > > **Weakness 4**
> > > >
> > > > > Minimal technical contribution: I’m impressed that the authors brought together many different techniques and got them to actually work on a real robot. However, this style of paper only makes a strong publication if the results are surprisingly strong. Given the previous works mentioned above, I do not believe the paper passes this bar in its current form, making the limited technical novelty an additional weak point.
> > > >
> > > >
> > > > While we agree that the individual components of our system build on established ideas, we believe the **effective integration of these elements into a robust, general-purpose pipeline for fully offline MBRL on real robots** constitutes a meaningful contribution. Our intention is not to introduce algorithmic novelty for its own sake, but to address an important and underexplored practical challenge: **how to make offline MBRL work safely and reliably in real robotic systems**.
> > > >
> > > > This framing is consistent with the papers cited by the reviewer. For example, *Smith et al. (RSS 2023)* explicitly acknowledge in Section III that *“our algorithmic framework is not novel… We emphasize that our result is not enabled so much by any one algorithmic component… but rather careful implementation and task setup.”* Likewise, the key contributions in *Levy et al. (CoRL 2024)* and *Yang et al. (CoRL 2020)* do not stem from novel algorithms, but from **establishing principled recipes for deploying existing MBRL or RL techniques on quadrupeds**, despite limited simulators and noisy real-world data. These papers are valued precisely because they show how to make real-world learning work in practice.
> > > >
> > > > Our contribution follows this same philosophy but tackles a **strictly more constrained and less-explored setting**. As detailed in **Sections A.5 and A.6**, fully offline MBRL introduces unique challenges: no corrective real-world interaction, substantially amplified distribution shift, compounding long-horizon model errors, and a lack of guidance on how uncertainty should be structured, penalized, and propagated over hundreds of steps. Despite extensive offline RL research in controlled simulators, we could not find prior work that provides a principled guideline for deploying offline RL—or uncertainty-aware long-horizon world models—on real robotic hardware.
> > > >
> > > > Therefore, the contribution of our paper lies not in proposing a new algorithmic building block, but in **identifying, analyzing, and addressing the design decisions that actually make offline MBRL viable for real robots**, and in demonstrating stable real-world deployment under the strict no-interaction constraint. We believe this fills an important gap between offline RL theory and real-world robotics practice, and we hope the community will find this effort as valuable as similar systems-focused contributions in the cited works.
> > > >
> > > >
> > > > **Question 1**
> > > >
> > > > > Why is offline MBRL the correct approach for real word learning for the tasks in the paper? Given that the current results do not “move the needle” in terms of capabilities, I believe this point needs to be thoroughly defended.
> > > >
> > > >
> > > > We appreciate the reviewer’s question and would like to clarify that our paper does **not** argue that offline MBRL is the universally best approach for real-world robot learning. As discussed in **Weakness 3**, if one *has access to online interaction with the robot*, then there is little reason to prefer offline RL—online reinforcement learning or online model-based control would dominate in terms of achievable capabilities.
> > > >
> > > > Our work is scoped around a **different and practically relevant setting**: situations where only a **fixed, pre-collected dataset** is available, and **no further environment interaction is possible or allowed**. This constraint arises in many robotics scenarios where safety, cost, hardware wear, operational interruptions, or deployment requirements prevent iterative data collection. In such settings, offline MBRL becomes the **only viable paradigm** capable of exploiting existing data while still enabling improvement beyond the behavior that generated it.
> > > >
> > > > Within this offline constraint, our goal is therefore not to claim that offline MBRL is the best general-purpose method for robot learning, but rather to **identify a principled and practical pipeline that maximizes the achievable performance delta from a fixed offline dataset**. The contribution of the paper is precisely to understand what components—long-horizon uncertainty propagation, uncertainty-aware policy optimization, and careful rollout design—are necessary to make offline MBRL work in real hardware conditions, a question that is currently underexplored.
> > > >
> > > > In short, offline MBRL is the correct approach **for the problem setting studied in this paper**—a **fully offline** scenario with **no** opportunity for further data collection—and our objective is to provide a robust and well-validated recipe for performing reliable policy learning under these strict constraints.

---

> > > > > ### Author Response · Authors · 2025-11-20
> > > > >
> > > > > **Question 2**
> > > > >
> > > > > > Can the given approach succeed when optimal data is not available?
> > > > >
> > > > > Yes, the approach is explicitly designed to succeed **when optimal data is not available**, and our experiments directly demonstrate this. As discussed in **Weakness 3**, the real-world dataset used in **Table 1** is collected by a **heavily randomized and clearly suboptimal policy**, which achieves only **0.79** normalized performance on hardware. Despite this, our offline MBRL pipeline attains **0.91**, substantially surpassing the behavior that generated the data. This is a core objective of offline RL: to extract policies that outperform the dataset, even when that dataset is far from optimal.
> > > > >
> > > > > More generally, offline MBRL—in contrast to behavior cloning or supervised imitation—does not require expert data. The uncertainty-aware world model and policy optimization components are specifically intended to handle **biased, noisy, and non-expert datasets**, which are the norm in real robotic systems.
> > > > >
> > > > > Thus, not only can our approach succeed without optimal data—**it is explicitly validated in that setting**, and its strongest real-world result comes from distilling a better policy from a clearly suboptimal dataset.

---

> ### Comment · Reviewer_uffN · 2025-11-25
> **Primary Concerns Not Addressed**
>
> I thank the authors for their detailed response. However, I feel that these responses are sidestepping my primary concerns. To be extremely clear about what I need to see:
>
>
> 1. I’m claiming that the fully offline setting is $\textbf{artificial}$, given that prior approaches have demonstrated significant improvement is possible with small amounts of real-world data and batch offline approaches. Fundamentally, if I wanted to deploy in the real world, why wouldn’t I iteratively improve performance with multiple rounds of offline learning? Given the limited technical novelty, this point needs to be defended carefully.
>
> 2. To my understanding, the paper has demonstrated that a) the method roughly matches the performance of baselines in simulation when optimal data is unavailable (Figure 4) and b) the method can improve performance compared to a data-collection policy using real-world data. This is not the same as showing that the method outperforms baselines when optimal data is unavailable, which is what I want to see. If I am misunderstanding the results in Figure 4, please clarify.

---

> > ### Author Response · Authors · 2025-11-25
> > **Concerns might come from potential misunderstanding of our scope and assumption**
> >
> > We thank the reviewer for the follow-up questions. We try our best to be clear about our arguments:
> >
> > > I’m claiming that the fully offline setting is $\textbf{artificial}$, given that prior approaches have demonstrated significant improvement is possible with small amounts of real-world data and batch offline approaches. Fundamentally, if I wanted to deploy in the real world, why wouldn’t I iteratively improve performance with multiple rounds of offline learning? Given the limited technical novelty, this point needs to be defended carefully.
> >
> > As we already stated in our response to **Question 1**, our paper **does not argue that offline MBRL is the best approach for real-world robot learning**. In fact, if one has access to online interaction with the robot, there is **no** reason to prefer offline RL, since fresh online data is undoubtedly better for improving the model/policy. We fully agree with the reviewer, and we are not disputing that.
> >
> > But is our setup "artificial"?
> > - **Yes**, in the sense that we could have collected more data online (which will for sure help). But the same logic applies to the whole field of simulation and robot learning: one can always collect more data with their simulators or robots. But then **why do people still do research on offline RL at all**?
> > - **No**, in the sense that online rollout can suffer from safety or certification limits, hardware access windows, cost/wear constraints, and operational downtime. In our case, the robot can **fall a lot, which is not acceptable for hardware safety**. In this case, **even if we could collect data online, the uncertainty penalization and the policy conservativeness facilitated by our approach are still required to reduce such online risk**.
> >
> > Overall, this paper does not argue that offline is better than online. Posing ourselves in a fully offline setup is a research assumption that defines the scope of this paper, while similar uncertainty penalization and a safe exploration philosophy also apply to online setups. Our study found that many existing methods were proposed to solve these "artificial" scenarios and were validated on "artificial" benchmarks (e.g., simulators that could have provided more online data), but not successfully on "artificial" real systems. Our work is thus to understand what makes it so challenging under the same assumption.
> >
> > > To my understanding, the paper has demonstrated that a) the method roughly matches the performance of baselines in simulation when optimal data is unavailable (Figure 4) and b) the method can improve performance compared to a data-collection policy using real-world data. This is not the same as showing that the method outperforms baselines when optimal data is unavailable, which is what I want to see. If I am misunderstanding the results in Figure 4, please clarify.
> >
> > We respectfully point out that the results in fig. 4 are misunderstood.
> >
> > a) Our method (**MOPO-PPO**) outperforms all the baselines when for **Expert** and **Mixed** datasets, and in **Mixed** there is no optimal data (in fact, it is worse when only optimal data is available). Our method (**MOPO-PPO**) and **MOPO** are built on the same concept of uncertainty penalization during policy training, so it is expected that they perform similarly across most tasks. The **real advanced performance** of **MOPO-PPO** in fig. 4 is attributed to **its compatibility with a better policy optimization method (PPO)**, which would not have been possible without the specific long-horizon uncertainty quantification technique introduced in RWM-O.
> >
> > **In short**, fig.4 shows: in cases where MOPO can solve the task, MOPO-PPO can also; but in cases where MOPO and other offline RL methods fail **due to their constraints on the policy optimization methods**, MOPO-PPO outperforms with its PPO compatibility, whose need for stable long-horizon autoregressive rollouts is only enabled by RWM-O. When faced with lower-quality data (**Random** and **Medium**), MOPO-PPO is not doing worse.
> >
> > b) **MOPO-PPO** can already improve performance compared to a data-collection policy **without any real data** (first column in Table 1) purely due to the policy improvement in offline RL. This improvement can be **further amplified** by providing an **appropriate** amount of **real data** (second column in Table 1).
> >
> > When faced with lower-quality data (**Random** and **Medium**), MOPO-PPO is not meant to outperform the baselines, as its advantage lies rather in policy optimization, which only starts to matter when higher-quality data is available, but not necessarily optimal (as evidenced in the **Mixed** datasets).
> >
> > We hope our answer clarifies our arguments and addresses the concerns the reviewer raised.

---

### Author Response · Authors · 2025-11-20
**General Response**

We would like to thank all reviewers for their valuable feedback. Here we summarize the major changes we made based on the feedback we received from the reviewers:

- We adjusted **Sections 1, A.5 and A.6** to include discussion and citation on suggested related work. *(Reviewers uffN, oixy, iN9s)*
- We added **Sec. A.6** to address ambiguity between online vs offline reinforcement learning and highlight the motivation and challenges in our fully offline setup. *(Reviewer uffN)*
- We adjusted **Sec. A.5** to detail the distinctive challenges and constraints we face when deploying offline MBRL algorithms to real hardware. *(Reviewers uffN, oixy, iN9s)*
- We also removed our initial strong phrasing from the abstract and revised the introduction to describe our contribution more precisely. *(Reviewer oixy)*
- We explained why many existing offline MBRL frameworks cannot be immediately applied to real-world robotics in **Sec. A.5**. *(Reviewers uffN, oixy, iN9s)*
- We added additional experiments using COMBO in **Sec 5.3** and updated **Fig. 4**. *(Reviewers oixy, iN9s)*
- We also corrected minor typos in the main text and the appendix.


In addition to the changes on the paper text, we uploaded additional videos on [our website](https://sites.google.com/view/iclr2026-rwm-o/home) addressing the following concerns:

- We added **additional videos of experiment results** to compare our method with state-of-the-art online model-free simulator-based methods. *(Reviewers uffN, iN9s)*


We uploaded the updated main paper and supplementary text. In the supplementary ZIP file, we provide `diff.pdf` to highlight the text changes we made. We hope the additional improvements could lead to improved paper quality. And we sincerely thank all reviewers for reviewing these latest changes.

---

### Meta-Review · Area_Chair_mX5G · 2026-01-07

**Summary:**

This paper developed a new framework of Offline Robotics World Models that extends the prior study of Robotics World Models to the offline setting, by incorporating uncertainty regularizations to the predictions of the dynamics model. The writing and presentation were overall clear, and the reviewers all agreed on the thoroughness of the ablations and engineering efforts to get the offline MBRL stack to run on hardware. However, there were also some common concerns on the limited conceptual and technical novelties, the missing and insufficiency of relevant baselines, as well as practical motivations. The rebuttal helped address some of the concerns, but not thoroughly. I recommend that the authors incorporate the feedback in preparing the next version of the paper.

**Reviewer Concerns:**

The concerns about the limited scope of robotic tasks, interpretation issues, the overly-strong claims about novelty compared to related work have been mostly addressed by the rebuttal. However, there are still outstanding concerns regarding the novelty, motivation, thoroughness of the experiments, and venue-fitness.

**Reviewer Scores:**

Reviewer oixy is likely to increase the score slightly, as the changes they requested, regarding title/abstract, are relatively easy to make. It was not clear if other reviewers will significantly change their scores.

---

### Decision · Program_Chairs · 2026-01-26

Reject